



# Interannual variability of Sea Surface Salinity in North-Eastern Tropical Atlantic: influence of freshwater fluxes

Clovis Thouvenin-Masson[1], Jacqueline Boutin[1], Vincent Échevin[1], Alban Lazar[1], Jean-Luc Vergely [2]

[1] Sorbonne Université, LOCEAN-IPSL, CNRS/IRD/MNHN, Paris, France
[2]ACRI-st, Guillancourt, France

*Correspondence to*: Clovis Thouvenin-Masson (clovis.thouvenin-masson@locean.ipsl.fr)  and  Jacqueline  Boutin (jb@locean.ipsl.fr)

**Abstract:**

In tropical regions, the fresh water flux entering the ocean originates primarily from precipitation and, to a lesser extent when
considering basin scale averages, from continental rivers. Nevertheless, at regional scale, river flows can have a significant impact on the surface ocean dynamics. Riverine fresh water modifies salinity, and therefore density, stratification and circulation. With its particular coastline, relatively high cumulative river discharge, and the vicinity of Inter Tropical Convergence Zone (ITCZ), the eastern Southern North Tropical Atlantic (e-SNTA) region off Northwest Africa is a particularly interesting location to study the linkage between precipitations, river outflows and Sea Surface Salinity (SSS).
Here we focus on the regional e-SNTA SSS seasonal cycle and interannual variability. We quantify the impact of river runoff and precipitation on SSS by means of regional simulations forced by different interannual and climatological river runoffs and precipitation products. The simulated SSS are compared with the Climate Change Initiative (CCI) SSS, *in situ* SSS from Argo, ships and a coastal mooring, and the GLORYS reanalysis SSS. The analysis of the salinity balance in the mixed layer is conducted to explore the dynamics influencing the SSS variability. Overall, the simulations reproduce well the seasonal cycle
and interannual variability despite a positive mean model bias north of 15N. The seasonal cycle is impacted by the phasing of the different runoff products. The mixed layer SSS decrease during the rainy season is mainly driven by precipitation followed by runoff by means of horizontal advection and partly compensated by vertical mixing. In terms of interannual anomalies, river runoffs have a more direct impact on SSS than precipitation. This study highlights the importance of properly constraining river runoff and precipitation to simulate realistic SSS, and the importance of observing SSS in coastal regions to validate such
constraints.

## 1 Introduction

The upper layer of the ocean is where exchanges between the ocean and the atmosphere take place. Forcings (e.g., wind, waves) generate turbulence and tend to create a mixed layer from a few meters to hundreds of meters thick, with homogeneous characteristics (e.g., temperature and salinity), and whose bottom is characterized by a marked density gradient, the pycnocline.
This layer is where the various flows exogenous to the ocean take place such as precipitation, or river discharge. These water flows impact the ocean in different ways: they make the density of the surface waters to decrease, hence increase the gradient



between the surface and subsurface waters. Freshwater flows can generate significant salinity gradients within the mixed layer (Mignot et al. 2007), leading to the formation of intermediate layers known as barrier layers. The latter isolates the surface layer from the deep ocean, inhibiting heat exchange between the ocean surface and subsurface (Vialard and Delecluse 1998).

Such ocean-atmosphere interactions make the mixed layer an ideal place to observe the water cycle, the generation of water masses and their evolution.

Since the 1980s, the quality and availability of *in situ* river discharge measurements have declined due to a lack of funding and to an unwillingness by states to share these data with the general public (Chandanpurkar et al. 2017, Durand et al. 2019). As a result, current Ocean General Circulation Model (OGCMs) such as used to generate Mercator's GLORYS reanalysis

(Lellouche et al. 2021) typically utilize climatological river discharge products such as Dai et al. (2009), which have not been updated for more than a decade. However, it has been shown that river discharges tend to vary strongly interannually. Gévaudan et al. (2022) found that the Amazon River runoff anomalies can reach values of the order of 50 000 $m^3s^{-1}$ (25 % of the climatological value) and that these anomalies have a significant influence on the salinity of the tropical Atlantic Ocean. Chandanpurkar et al. (2022) studied the influence of river discharge interannual variability on salinity at the mouths of the

world's major rivers. They found that river discharge interannual variability is responsible for a standard deviation of 1.3 to 3 pss of salinity, and that models that take the interannual variation of river discharge into account simulate SSS that are closer to satellite observations. At the scale of the global ocean, a recent study (Fournier et al. 2023) demonstrates that SSS variability, averaged in estuarine regions of major river plumes, is strongly correlated with the global water cycle variability, particularly in relation to the ENSO phenomenon.

In this paper, we focus on the eastern Southern-North Tropical Atlantic region (e-SNTA, 10°N-17°N/20°W-12°W; Figure 1), a region highly subject to freshwater forcing. We analyze the origin of the observed SSS anomalies, and the importance of each forcing term on these anomalies. The North-West African region (Figure 1) is the site of significant river outflows, resulting from rainfall over the high mountain plateaus of Guinea. There is a geographical disparity in the river flows: to the north of Dakar (14.7°N), only the Senegal River has an average outflow of over 500 $m^3s^{-1}$ (Roudier et al. 2014), whereas to

the south of Dakar, freshwater discharge takes place via a multitude of rivers along the coast, the most important of which being the Gambia River. River flow in e-SNTA is highly seasonal with rivers running almost dry during the boreal summer and peaking in autumn after the rainy season. While their interannual variations are not well known due to a lack of data, studies based on climate models (Ardoin-Bardin et al. 2009) predict a long-term decreasing trend for these flows by up to 27 % for Senegal river and up to 37 % for Gambia river in 2080.

Cumulating all the rivers discharges of Senegal and Guinea [12°N-17°N] leads to an average monthly outflow of ~30 000 $m^3s^{-1}$ at its annual maximum in September. In comparison, the largest Amazon outflow in May is 276 000 $m^3s^{-1}$, and the largest outflow of the Congo River is 56 000 $m^3s^{-1}$ in December (Wohl and Lininger 2022). However, e-SNTA region is of particular interest because it is subject to both river discharge and intense precipitation linked to the meridional displacements of the InterTropical Convergence Zone (ITCZ). Furthermore, this region hosts the thriving Senegalese coastal upwelling, a region

where human populations are highly dependent on small pelagic fisheries as a source of protein (Failler et al. 2014).




Concerning the impact of freshwater fluxes on salinity in e-SNTA region, only the seasonal variation has been studied by Camara et al. (2015) using the Nucleus for European Modelling of the Ocean (NEMO) ocean model. They found that runoffs and precipitations were the main contributors of the freshening in the e-SNTA, and that poleward advection of low salinity waters along the coasts was partly compensated by vertical diffusion of salinity.

In this context, the present study aims to complement the study by Camara et al. (2015) by quantifying the effect of interannual variation in forcings (e.g. runoffs) on surface and mixed-layer salinity in the e-SNTA. To this aim, the Coastal and Regional Ocean Community (CROCO) model is used and surface ocean dynamics are simulated using different configurations of climatological or interannual forcings. The model results are compared with Mercator's GLORYS reanalysis dataset, satellite and *in situ* SSS measurements (*e.g.*, merchant ships, Argo floats, buoys). The interannual variation in salinity in each

configuration is estimated, as well as the SSS physical drivers using a mixed-layer salinity balance following Camara et al. (2015).

Sect. 2 presents the data, and the methods used. Sect. 3 is dedicated to the results and includes a validation of the modeled SSS, an analysis of the observed and modeled SSS anomalies and a study of modeled SSS sensitivity to changes in freshwater flux forcings. Finally, a few points are raised for discussion and conclusions in sect. 4.


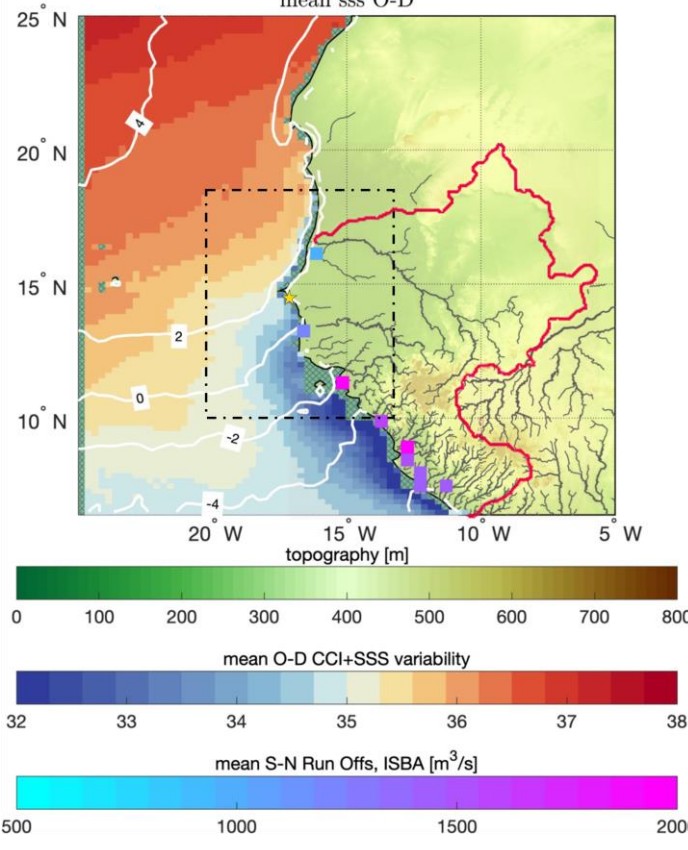



**Figure 1: satellite CCI SSS (color, in pss) averaged over October-November-December (over ocean); dashed white contours indicate the averaged ERA5 E-P (evaporation minus precipitation) rate (in mm/d) over this period. Topography (color scale, in meters) is shown on land. Along the coast, colored squares indicate averaged river outflows over September-October-November (color scale, in m³/s). The area delimited by the red line corresponds to the merged catchment areas of the rivers flowing into the area of study, extracted from the HydroSHEDS database (Hydrological data and maps based on SHuttle Elevation Derivatives). Black dotted box delimits the e-SNTA region. The yellow star represents the position of Melax buoy, moored at 30 m depth, ~20 km from the coast.**

## 2 Data & methods

### 2.1 The region of study: the Eastern-South North Tropical Atlantic (e-SNTA) [10°N 17°N, 20°W 5°W]

This region is identical to the one studied in Camara et al. (2015), for the purpose of comparing the results obtained. This region is strongly impacted by river water outflows, as it includes major rivers (Senegal, Gambia, Casamance, big and Little Scarcies; see Figure 1). In the following, we study the salinity and fresh water forcings variables averaged over this region.

### 2.2 SSS data

#### 2.2.1 Satellite maps

Three L-band radiometric satellite missions have measured SSS from space: SMOS (2010-present), SMAP (2015-present), and Aquarius (2012-2015). The version 3.2 of the SSS product generated as part of the Climate Change Initiative project (CCI) is used here, covering a period from 2010 to 2021 (Boutin 2021). These data are generated with a temporal optimal interpolation of the three satellites measurements, as described for version 2 in Boutin et al. (2021). Developments between version 2 and version 3 are described in detail in Thouvenin-Masson et al. (2022). SSS fields are available on a 25 km Equal-Area Scalable Earth (EASE2) and they are used here at a weekly temporal resolution. Due to the spatial resolution of satellite SSS measurements, data taken at less than ~40 km from the coast are flagged as they must be considered with caution due to land contamination (e.g. (Zine et al. 2008). This flag filtering is applied in the present study. In this satellite product, a correction is applied to remove the instantaneous effect of rain on the top surface satellite measurements to remain consistent with bulk salinity recorded by most *in situ* instruments (Supply et al. 2020).

#### 2.2.2 GLORYS reanalysis

The GLORYS12V1 product is a Copernicus Marine Environment Monitoring Service (CMEMS) global ocean eddy-resolving reanalysis available from 1993 to 2023. These reanalyses are based on the NEMO ocean model forced by ERA5 data and by climatological river runoffs at the surface. Satellite sea level anomalies, Sea Surface Temperature (SST), sea ice concentration, *in situ* temperature, and salinity vertical profiles (but not satellite SSS) are assimilated using a reduced-order



Kalman filter derived from a singular evolutive extended Kalman (SEEK) filter with a three-dimensional multivariate background error covariance matrix and a 7 day assimilation cycle (Lellouche et al. 2018, Lellouche et al. 2021). Model reanalysis output is available at daily temporal resolution on a regular 1/12° grid for 50 vertical levels. See Lellouche et al. (2021) for a complete description of the model.

### 115 **2.2.3 Salinity *in situ* data**

*In situ* data are used to evaluate the CROCO simulations. This involves thermosalinographs (TSG) installed on merchant and research vessels, measurements from Argo floats near the surface, and the Melax mooring measurements.

- **TSG**

The delayed mode data set from Laboratoire d'Etudes en Géophysique et Océanographie Spatiales (TSG-LEGOS-DM) is used. It is derived from voluntarily observing ships, collected, validated, archived and made freely available by the French Sea Surface Salinity Observation Service (Alory et al. 2015). Adjusted values, when available, and TSG data with quality flags = 1 and 2 ('good' or 'probably good') only are selected.

- **Argo profilers**

The Argo project is a set of about 4000 profilers moving in the global ocean. These instruments provide around 100 000 temperature and salinity measurements annually over the global ocean, and with an average spacing of 3 degrees between measurements (Argo 2023). These data are collected and made freely available by the international Argo project and the national programs that contribute to it. The Argo SSS data gathered in the Salinity-Pilot Exploitation Platform (Pi-Mep)

database (Guimbard et al. 2021) is used. In this database Argo data from the Global Data Assembly Centre (GDAC) database (Argo 2023) with a quality index of 1 or 2 are selected. Argo measurements between 10 m and 0 m depth are considered as surface data (most of the Argo data resulting from this selection are taken at a depth of about 5 m).

- **Melax**

The Melax mooring is equipped with oceanographic and atmospheric sensors. Moored at 36 m depth, it is located at 30 km from the coast. It measures the physical and biogeochemical parameters over the Senegalese shelf [ 14°20' N,17°14' W], south of the city of Dakar (Tall et al. 2021). The mooring captured surface salinity almost continuously from mid-February 2015 to august 2016.





### 2.2.4 Regional simulations: the CROCO model

The ocean model CROCO (https://www.CROCO-ocean.org/;(Hilt et al. 2020) is used to simulate salinity variations in the e-SNTA region. CROCO has vertical sigma coordinates, which are well suited for coastal studies. The slow baroclinic mode and the fast barotropic mode are computed separately (Shchepetkin and McWilliams 2009), improving the consistency, accuracy, and stability of the simulations. High-order numerical schemes enable the representation of small-scale structures such as mesoscale eddies and filaments. The AGRIF (adaptive refinement of the horizontal grid ; (Debreu et al. 2008) module is utilized, enabling the embedment of a sub-domain in which small-scales are more finely resolved. In the configuration used here, the parent grid covering [7°N-35°N; 30°W-10°W] has a resolution of 10 km, and the child grid used in the Senegal region [12°N-18°N; 20°W-15°W] has a resolution of 2 km. More details on the model configuration can be found in Ndoye et al. (2018). Daily outputs from the MERCATOR model output at 1/12° resolution (GLOBAL-ANALYSISFORECAST-PHY-001-024; downloaded from http://marine.copernicus.eu/) are used to force physical properties (temperature, salinity, velocity and sea level) at the open boundary conditions (OBC) of the parent grid.

Hourly atmospheric forcings (air temperature, relative humidity, 10 m wind, radiative fluxes) from the ERA5 reanalysis (see below) are used in all simulations. In the present study, the model was run with two different precipitation datasets and two different river runoff products, which are detailed in the following. No surface salinity restoring to climatological observations (*e.g.,* Ndoye et al. (2018)) was used.

### 2.2.5 Precipitation forcings

- **ERA-5**

ERA5 is a reanalysis produced by the European Centre for Medium-Range Weather Forecasts (ECMWF), which provides comprehensive modeling of atmospheric, continental surface and ocean wave variables (Hersbach et al. 2020). Based on the Cycle 41r2 Integrated Forecast System (IFS), ERA5 hourly fields have a horizontal resolution of 31 km over the period 1950-2023.

The precipitation value used in the CROCO simulations is composed of the convective precipitation field (cp) produced by the IFS convection scheme, which represents precipitation at sub-grid scales, and the stratiform precipitation field (sp) produced by the IFS cloud model, which represents the formation and dissipation of clouds and large-scale precipitation due to changes in atmospheric variables such as pressure, temperature and humidity.

- **IMERG**

Integrated Multi-satellitE Retrievals for Global Precipitation Measurement (IMERG) is a rain rate product based on satellite precipitation measurements. It combines information from the Global Precipitation Measurement (GPM) satellite constellation



with infrared (IR) satellite data taken by geostationary satellites to estimate precipitation over the majority of the Earth's surface at a frequency of 30 minutes. The algorithm is based on the Climate Prediction Center Morphing (CMORPH; (Joyce et al. 2004) method, and takes advantage of the high repetition rate of IR satellites to track the movement of less frequent but more

accurate microwave and radar-detected rainfall cells.

Over the ocean, these two precipitation products are consistent in terms of mean values and have similar climatologies and anomalies (Figure 5c, Appendix B), after integration over the e-SNTA. Nevertheless, IMERG rain rates are more variable locally and extend over a larger range of values than ERA5 rain rates (see Appendix A, Fig. A.1 and A.3).


### 2.2.6 Runoffs forcings

River discharges estimated by the GloFAS and ISBA-CTRIP models are used to force CROCO. The characteristics of the forcings are listed below:

–  **GloFAS**

The Global Flood Awareness System (GloFAS; http://www.globalfloods.eu/; (Harrigan et al. 2020) is one of the components of the Copernicus Emergency Management Service (CEMS). This system is designed to help prevent flooding on a global scale, notably by providing water level forecasts for river basins. It is based on satellite data, on soil temperature and humidity, on precipitation from ERA5, and on *in situ* data. These data are integrated into the Hydrology in the Tiled ECMWF Scheme for Surface Exchanges over Land (HTESSEL) continental surface model, which is part of the ECMWF's integrated forecasting

system (IFS 41r2) via a terrestrial data assimilation system explained in de Rosnay et al. (2012). The resulting runoff is then integrated into the LISFLOOD runoff routing model.

–  **ISBA-CTRIP**

ISBA-CTRIP river discharge estimation is used in this study. ISBA-CTRIP combines two models:

the Interaction Soil-Biosphere-Atmosphere (ISBA; https://www.umr-cnrm.fr/isbadoc/model.html) hydrological model developed by the Centre National de Recherches Météorologiques (CNRM) within the framework of the IPCC (see Decharme et al. (2019) for a full description of the model) and the CTRIP (CNRM version of Total Runoffs Integrating Pathway) model, which is an improved version of the TRIP model used to simulate river runoff to the ocean from the total runoff calculated by ISBA. In the configuration used here this model uses Tier-2 Water Resources Re-analysis precipitation at 0.25° resolution

(WRR2) from the E2O project as forcing. The E2O data set is directly based on the 3-hourly ERA-Interim reanalysis (https://www.ecmwf.int/en/forecasts/datasets/reanalysis-datasets/era-interim) over the 1979-2014 period. Precipitations have been hybridized with observations using the Multi-Source Weighted-Ensemble Precipitation (MSWEP; http://www.gloh2o.org) data set (Beck et al. 2017).



GloFAS and ISBA runoffs, after summing the individual outflows for the region studied, have similar climatologies (maximum difference of $1.10^8 \, m^3/d$, see Appendix B). The simulated river runoffs exhibit strong interannual anomalies in this area (Fig. 5c). These river runoff anomalies are highly correlated with monsoon anomalies (Fig. 8) and can reach 11 000 $m^3 s^{-1}$, i.e., almost 30 % of the seasonal variation (Appendix B). These interannual anomalies (see sect. 3.3 below) are sometimes of opposite signs between the two products, with differences reaching $12.10^8 \, m^3/d$ (Figure 5c). These differences and their origins

are discussed in section 4.

**2.3 Characteristics of the CROCO simulations**


In order to estimate the relative importance of interannual variations in each of the freshwater fluxes forcings, five CROCO simulations are performed with different rain rate and river runoffs forcings. The other hourly forcing terms (air temperature, wind, radiative flux, etc.) are kept identical for all simulations. Three simulations are forced with synoptic freshwater flux forcings, including interannual variations. These simulations are called CROCOglofas, CROCOisba and CROCOimerg. Two

simulations are forced with climatological rain rates or climatological river outflows respectively. They are named CROCOprclm and CROCOroclm. The forcings of each simulation are summarized in Table 1.

**Table 1: list of simulations studied and their freshwater flux forcings.**

| Name | Precipitation | River discharge |
|---|---|---|
| **Interannual simulations** | | |
| CROCOglofas | ERA5 hourly | GloFAS daily |
| CROCOisba | ERA5 hourly | ISBA daily |
| CROCOimerg | IMERG hourly | ISBA daily |
| **Climatological simulations** | | |
| CROCOroclm | ERA5 hourly | GloFAS climatology |
| CROCOprclm | IMERG climatology | ISBA daily |




## 2.4 Analysis of the processes controlling the mixed-layer salinity budget

Diagnostics implemented in CROCO make it possible to isolate the various terms involved in the salinity balance of the mixed layer in order to identify the dynamical processes that modify salinity. In CROCO, the temporal variation of the salinity, $\partial_t S$,

is expressed as follows, for each layer of the water column:

$$\partial_t S = -\partial_x(uS) - \partial_y(vS) - \partial_z(wS) + \partial_z(K_z \partial_z S) \tag{1}$$

With t the time dimension, x, y, z the zonal, meridional and vertical dimensions respectively, u, v, w the current on the x, y, and z dimensions respectively, S the salinity, $K_z$ the vertical diffusion coefficient.

The boundary conditions on salinity fluxes are:

- At surface $(z = 0)$: $K_z \partial_z S = SSS(E - P)/\rho_0$

- At ocean bottom $(z = -H)$: $K_z \partial_z S = 0$ (no exchanges through the bottom)

Over the mixed layer of height (h), the salinity budget is computed as follows:

$$(1/h) \cdot \int_{-h}^{0} (\partial_t S) \cdot dz = (1/h) \cdot \int_{-h}^{0} \left( -\partial_x(uS) - \partial_y(vS) - \partial_z(wS) + \partial_z(K_z \partial_z S) \right) \cdot dz \tag{2}$$

Note $S_m$ the depth-averaged salinity in the mixed layer: $S_m = = 1/h. \int_{-h}^{0} S dz$

The left-hand side can be expressed as the sum of the time variation of $S_m$ and the entrainment term:

$$(1/h) \cdot \int_{-h}^{0} (\partial_t S) \cdot dz = \partial_t \left( (1/h) \cdot \int_{-h}^{0} S \cdot dz \right) + \partial_t h/h \cdot \left( 1/h \int_{-h}^{0} S \cdot dz - S(-h) \right)$$

$$\partial_t S_m + \partial_t h/h (S_m - S(-h))/h = -(1/h). \int_{-h}^{0} -\partial_x(uS) dz - (1/h) \int_{-h}^{0} \partial_y(vS) dz$$

$$-(1/h) \int_{-h}^{0} \partial_z(wS) dz + (1/h) \cdot \int_{-h}^{0} \partial_z(K_z \partial_z S) . dz \tag{3}$$

The last term (related to vertical diffusion) is equal to:

$$(1/h) \int_{-h}^{0} \partial_z(K_z \partial_z S) \cdot dz = (1/h) \cdot [K_z \partial_z S]_{-h}^{0} = (1/h) \cdot SSS(E - P)/\rho_0 - (1/h)[K_z \partial_z S]_{-h} \tag{4}$$

CROCO computes online (for each time step) $S_m$ and $\partial_t S$, so it can compute the entrainment term as a residual:

$$\partial_t h/h \cdot \left( 1/h \int_{-h}^{0} S dz - S(-h) \right) = (1/h) \cdot \int_{-h}^{0} (\partial_t S) \cdot dz - \partial_t \left( (1/h) \cdot \int_{-h}^{0} S dz \right) \tag{5}$$

Thus, the final equation is:



$$\underbrace{\partial_t S_m}_{rate} = -\underbrace{(1/h) \cdot \int_{-h}^{0} - \partial_x(uS)dz}_{zonal\ advection} - \underbrace{(1/h) \cdot \int_{-h}^{0} \partial_y(vS)dz}_{meridional\ advection} - \underbrace{(1/h) \cdot \int_{-h}^{0} \partial_z(wS)dz}_{vertical\ advection}$$

$$+ \underbrace{(1/h) \cdot SSS(E-P)/\rho_0}_{forcing} - \underbrace{(1/h)[K_z\,\partial_z S]_{-h}}_{vertical\ mixing} - \underbrace{\partial_t h\big(S_m - S(-h)\big)/h}_{entrainment} \qquad (6)$$

In this study, the mixing terms were found to be negligible compared to the other terms, and the horizontal and vertical advection terms were found to largely offset each other. Thus, in the following, the so-called advection term comprises the sum of the zonal, meridional and vertical advection terms. The runoff forcing is introduced via the zonal advection term $- \partial_x(uS)$ as rivers are introduced as westward zonal flows at different locations of the west african coastline.

**2.5 Comparison between SSS datasets and model variables**

**2.5.1 Calculation of lag-correlations**

In the following the spatial averages of the different variables linked to the salinity budget over the e-SNTA, and their relationships with each other are studied. We correlate forcing terms with salinities or other terms (such as wind intensity or MLD) using the determination coefficient ($r^2$). In order to identify cause-and-effect relationships that can take weeks to establish, the correlation maximum by allowing a time delay (up to ±90 days) between the variables is used.

**2.5.2 Comparison between SSS datasets**

-    **Colocation method between gridded SSS and Melax mooring or Argo floats SSS**

A spatial bilinear interpolation of the gridded product and a selection of the nearest neighbor in time are used to collocate products represented on a grid with local data from Argo floats or Melax buoy.

-    **Colocation method between gridded SSS and TSG SSS**

Given the high spatio-temporal sampling of TSG measurements, we smooth them with a gaussian window along the ship track to a resolution comparable to gridded datasets (e.g., model or satellite data type). The standard deviation of this filter is set to one quarter of the spatial resolution of the model grid (95 % of the weight in a radius of half the spatial resolution). The resulting smoothed TSG data are compared to the nearest corresponding satellite or model pixel in time.

Only the SSS in pixels that are common to modeled and satellite SSS are considered when comparing results obtained with both datasets.



**2.6 Climatological and interannual variability of the salinity budget**

The different variables linked to salinity budget are spatially averaged over the e-SNTA, and the resulting time series are analyzed from January 2010 to July 2019, which is the time period over which ISBA runoff was available at the time of this study. The averaged seasonal and interannual signals are then extracted from the original signal, as follows:

- Seasonal signal: a two-stage method is used to calculate a climatological seasonal variation. A daily climatology is first calculated by averaging data available on each day of the year between 2010 and 2019 (Figure 2). To eliminate short-term fluctuations, the daily climatology is then smoothed using a 1-month moving-average filter.

- Interannual signal: To remove the seasonal variations, the monthly climatology is subtracted from the original daily time series. A 3-month moving average is then applied to filter intraseasonal variability in order to focus on interannual variability at seasonal time scales.

**3. Results**

**3.1 Analysis and validation of the seasonal cycle of salinity**

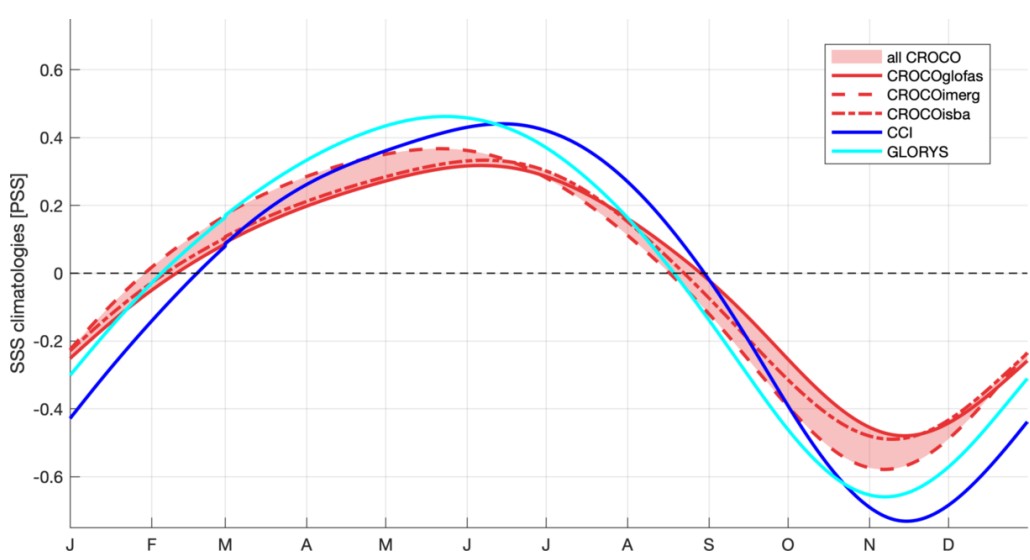

**Figure 2: Seasonal cycle of SSS in the e-SNTA region: CCI (blue), GLORYS (cyan) and various CROCO simulations. The red area represents the range between the minimum and maximum value of the simulated SSS.**

A study of the salinity balance resulting from the different CROCO simulations (see appendix B) was carried out in order to analyze the dynamical processes at the origin of the SSS seasonal cycle. The SSS climatological variations are governed by the effect of precipitation during the summer rainy season. Rainfall accumulates on the continent during this



period, generating intense river runoff 1-2 months later. The peak of river runoff is reached in September-October. This drives a continuous drop in salinity from the start of the rainy season onwards. These two effects also generate a strong vertical salinity gradient at the base of the mixed layer, and a decrease of the MLD. The ocean attenuates this freshwater flow through vertical advection (not shown) and entrainment of relatively fresh water towards the ocean interior. During the dry season

(January-May), the atmospheric forcing is slightly positive (due to evaporation being larger than precipitation) and associated with a thinner mixed layer. There is also a strong vertical inflow of salt in the mixed layer by advection (not shown), which corresponds to the coastal upwelling particularly marked in March. Relatively salty upwelled waters are then redistributed by horizontal advection, notably by westward Ekman transport. This analysis is in line with Camara et al. (2015).

305        Although similar in shape and of the same order of magnitude, the seasonal cycles shown in Figure 2 present notable differences: CROCO SSS have a seasonal cycle of smaller amplitude than that of CCI and GLORYS, and CCI SSS is in phase with CROCOglofas. On the other hand, CCI SSS lags CROCO SSS when forced by ISBA runoffs (CROCOimerg and CROCOisba) and GLORYS SSS by ~2 weeks.  Between the CROCO simulations, there is a difference of the order of 0.1 pss in amplitude, which may stem from a difference in the amplitude of the seasonal cycle of the precipitation products that were

used, the amplitude of IMERG seasonal cycle being $3.10^8$ m³/day larger than the one of ERA5 (Appendix B, Fig. B.1.b).

### 3.2 Evaluation of the CROCO simulations using *in situ* measurements

#### 3.2.1 Coastal SSS from the Melax mooring off southern Senegal

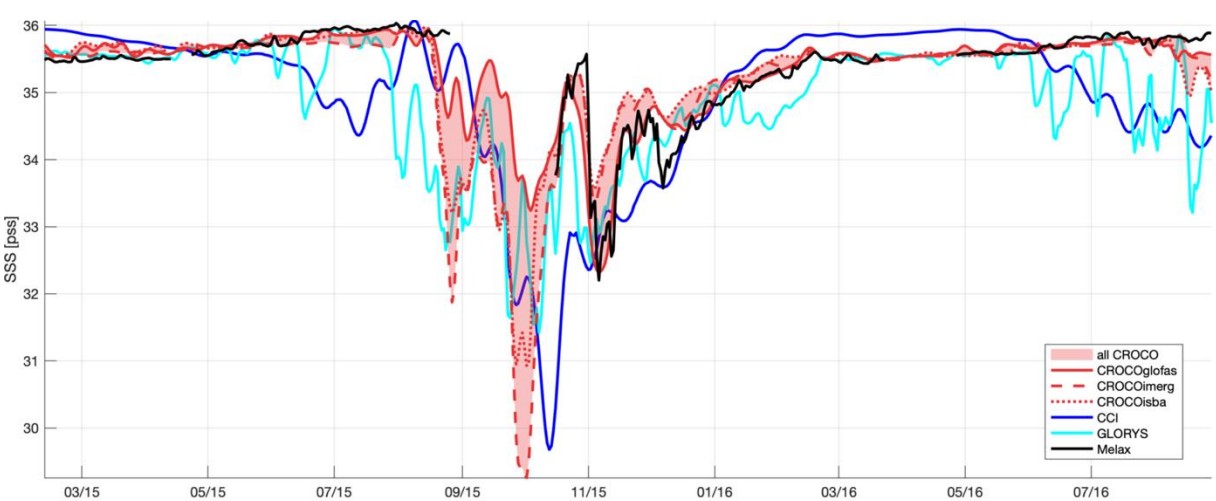

**Figure 3: Surface salinity from the Melax mooring (black line) between March 2015 and August 2016. The CROCO simulations are**
**marked by red lines. Red shading shows the range between the maximum and minimum simulated values. Satellite data is shown in blue, GLORYS data is shown in cyan. Both time series are collocated at the Melax mooring. Notice that the nearest satellite data pixel is further than 30 km apart from the mooring position and further than 50 km apart from the coast.**



Salinity measurements at the Melax mooring provide a useful time series for the evaluation of the simulation near the Senegalese coast. During its first two years of deployment, the mooring recorded an almost continuous time serie, with a SSS

oscillation of ~2-3 pss amplitude in November 2015 (Figure 3). CROCO SSS agree well with *in situ* SSS, during the dry seasons (March-September 2015 and March-July 2016) and at the time of the very strong oscillation in November. GLORYS also gives consistent results, but underestimates the amplitude of the first observed oscillation of SSS (Mid October – early November 2015). GLORYS SSS is also highly oscillatory and too low over the periods when *in situ* SSS is stable (before September 2015 and after May 2016). The CCI SSS is further from *in situ* SSS, which is expected as the pixel collocated with

the mooring is 30 km offshore of the mooring (and 55 km from coast), due to the application of the coastal flag and the land contamination close to the coast. In addition, in such a coastal area with unresolved satellite SSS variability, an error arises from the sampling difference between a pointwise *in situ* measurements and a satellite measurement integrated over ~50km (Thouvenin-Masson, et al. 2022), which is greater than the GLORYS and CROCO horizontal resolutions.

The salinity balance is used to explain the origin of the strong oscillation detected in mid-2015 (see Appendix C). Freshening

is initiated in August 2015 by an event of intense precipitation, and amplified by advection of freshwater from the coastal regions south of the mooring which collect a strong river runoff until November 2015. The observed oscillation in mid-November 2015 is caused by an oscillation of the zonal advection term, leading to an intensified westward (eastward) transport of relatively low (high) salinity waters (not shown).

**3.2.2 Argo and TSG**

Although most of the *in situ* measurements are taken at a depth between 5 and 10 meters, it has been chosen to compare them with the salinity in the top layer of the model, in order to be able to analyze these validations in the light of comparisons with satellite measurements taken at the first centimeter of the ocean (note also that there are no strong vertical salinity gradients in the top 5 m of the model water column).

Of the three types of gridded products, satellite observations are the closest to *in situ* data, with $r^2$ values of 0.94 and 0.89 for comparisons with Argo and TSG data respectively (Table 2). With the exception of TSG measurements taken very close to the coast where the satellite data are slightly overestimated, the differences observed rarely exceed 0.2 pss in absolute value (Figure 4a.e.).



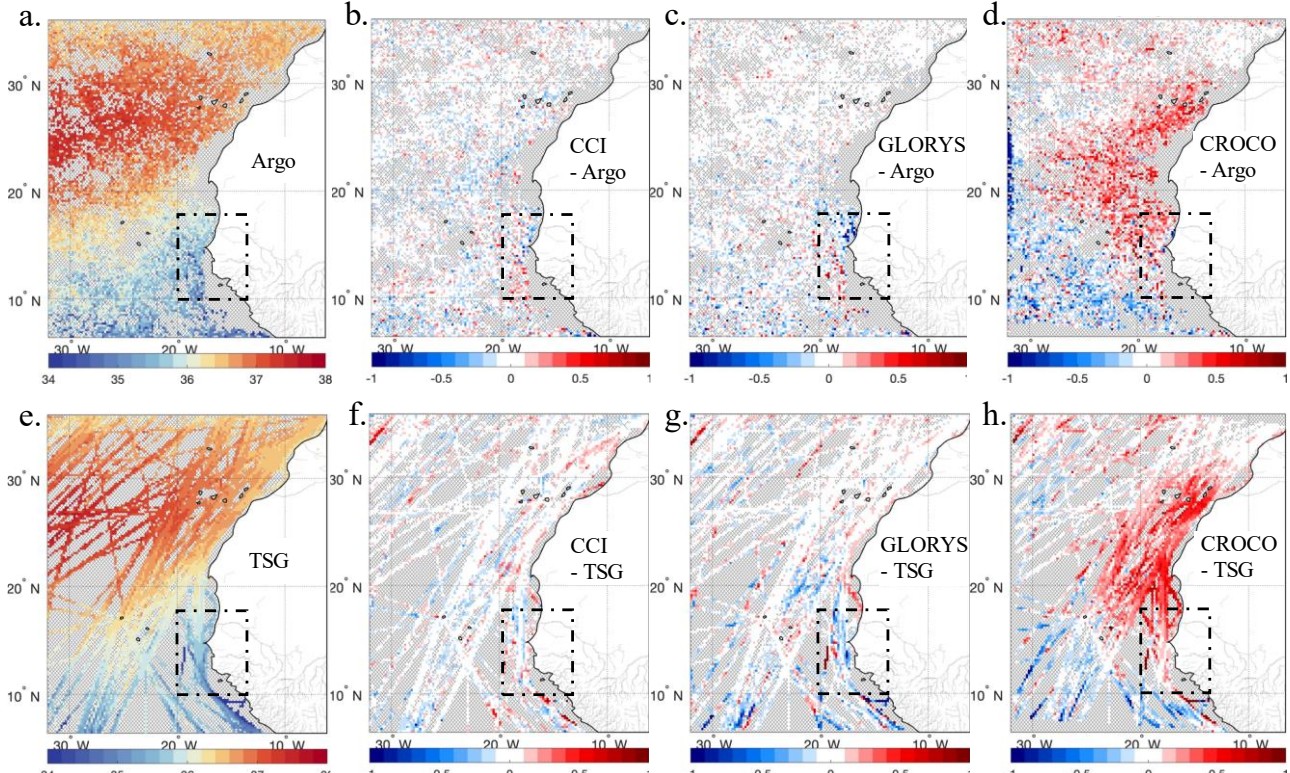


**Figure 4:(top) Argo SSS (a) used as reference. Difference between Argo SSS and various SSS fields (CCI (b), GLORYS (c) and CROCOimerg (d)). (bottom) TSG SSS (e) used as reference. Difference between TSG SSS and various SSS fields (CCI (f), GLORYS (g) and CROCOimerg (h)). Comparisons were averaged over 0.2° boxes to ease visualization.**

Since the GLORYS reanalysis assimilates Argo data, the statistics of the comparisons with this dataset are very good, as
expected. The $r^2$ values are 0.91 when comparing to Argo, and 0.86 when comparing to TSG, only slightly lower than those obtained with CCI (Table 2). However, there is a negative bias with respect to Argo and TSG measurements taken on the continental shelf at the mouth of the Senegal River (16°N). These Argo data were taken at the end of 2012, which corresponds to a period when river outflows were particularly high (Fig. 5c). It is therefore likely that these differences of around -0.8 to -1 pss can be explained by the use of climatological runoff in GLORYS (Figure 4c.g.).

When CROCO SSS are compared with *in situ* SSS, a significant bias of up to +0.5 pss is observed, which is fairly systematic near the coast and north of 14.7°N. This positive bias is observed with respect to both Argo and TSG data and with all CROCO simulations, thus seems fairly robust. South of 14.7°N and far from the coast (30°W-20°W), a negative bias of the order of -0.2 pss is observed, while the few TSG data available on the continental shelf show a positive bias (Figure 4d.h). There is a stronger bias in CROCOprclm (Table 2), suggesting that the climatological precipitation field strongly attenuates the effect of
rainfall on SSS. Comparison statistics between the other simulations and *in situ* data are very close (with maximum absolute differences in terms of $r^2$ of the order of 0.01), with slightly better results for CROCOimerg.



The seasonal variability of the CROCO SSS bias with respect to CCI shows similar patterns (positive bias near the north Senegalese – Mauritanian coast), regardless of the year (not shown). In the following, these simulations are therefore analyzed on a relative basis after removing the annual mean bias of SSS. The figures shown below are based on fields from which the mean SSS has been removed. The origin of the systematic SSS bias in CROCO is discussed in sect. 4.

The statistics calculated for the e-SNTA region are provided for reference in Table 2. There are significantly fewer co-located points in this region, lower dynamics of SSS and a higher proportion of points close to the coast compared to the global region. Consequently, the statistics are consistently less favorable, with $r^2$ values reaching 0.43 (0.67) compared to Argo floats (TSG).

**Table 2: Summary statistics of comparisons between various salinity products and in situ data over the global / e-SNTA areas (see maps in Fig. 4). "Std diff" stands for "standard difference".**

|  | Argo (17378 / 902 points) | | | TSG (133139 / 8033 points) | | |
|---|---|---|---|---|---|---|
|  | $r^2$ | Std diff | bias | $r^2$ | Std diff | bias |
| CCI | 0.94/0.70 | 0.16/0.30 | -0.01/0.01 | 0.89/0.65 | 0.20/0.37 | -0.01/-0.02 |
| GLORYS | 0.91/0.69 | 0.21/0.32 | -0.02/0.01 | 0.86/0.64 | 0.23/0.38 | -0.01/-0.02 |
| CROCOglofas | 0.81/0.43 | 0.30/0.45 | 0.07/0.19 | 0.77/0.65 | 0.30/0.36 | 0.14/0.14 |
| CROCOroclm | 0.80/0.25 | 0.31/0.62 | 0.06/0.14 | 0.77/0.66 | 0.30/0.35 | 0.14/0.13 |
| CROCOisba | 0.81/0.34 | 0.30/0.54 | 0.06/0.11 | 0.76/0.67 | 0.30/0.35 | 0.14/0.12 |
| CROCOimerg | 0.82/0.40 | 0.30/0.54 | 0.03/0.07 | 0.78/0.66 | 0.29/0.36 | 0.11/0.10 |
| CROCOprclm | 0.69/0.06 | 0.36/0.63 | 0.24/0.28 | 0.67/0.59 | 0.34/0.42 | 0.27/0.23 |



## 3.3 Origin of the main interannual variations

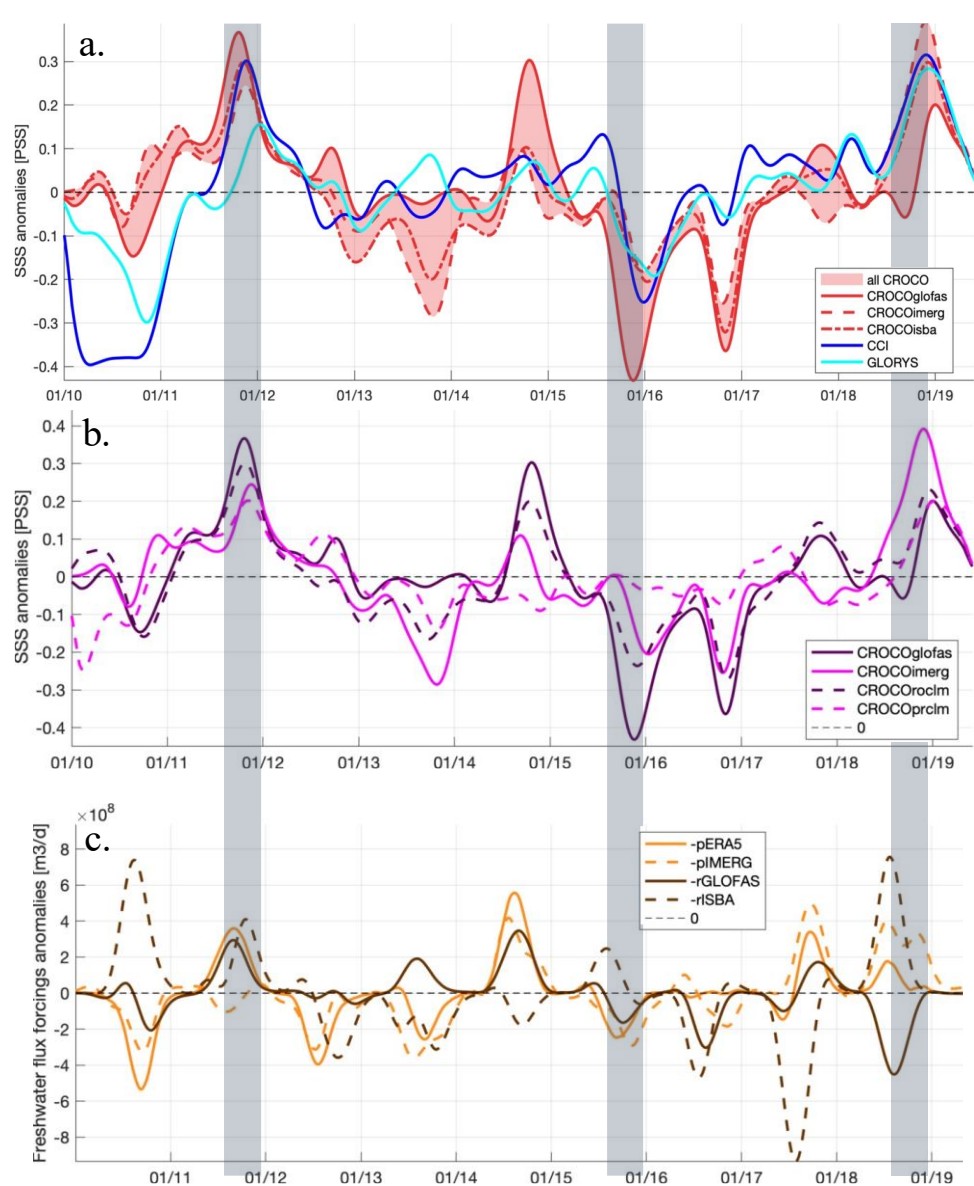

**Figure 5: Band-passed SSS anomalies for a) CCI (blue), GLORYS (cyan) and the three interannual CROCO simulations (red). The red shading represents the range between the minimum and maximum of the simulated SSS. b) SSS anomalies from the CROCO simulation with interannual runoffs (purple solid line) and climatological runoffs (purple dotted line). CROCO simulation SSS anomalies with interannual rain rate (magenta solid line) and climatological rain rate (magenta dotted line). c) Anomalies of the different forcing products, runoffs (brown, GLOFAS plain line, ISBA dashed line) and precipitation (orange, ERA5 plain line, IMERG dashed line). Note that the inverse of the freshwater flux is shown for easier comparison with SSS anomalies. Periods in grey shading correspond to large SSS anomalies for which a detailed analysis is given in section 3.3.**





The interannual anomalies calculated for the different CROCO simulations are now compared to the interannual anomalies of the CCI and GLORYS products, used as references given their good agreement with *in situ* data (Table 2).

Interannual variations of SSS in the e-SNTA are significant, oscillating between -0.4 pss and 0.4 pss and therefore of the same order of magnitude as the seasonal cycle (Figure 2). There is no long-term trend in the anomalies: for each year, anomalies are close to zero during the first half of the year (late winter-summer), and reach their extrema at the end of the year (fall-early

winter), lagging by a few months the anomalies of rainfall and runoff (Figure 5c).

Overall, the different SSS estimates are in relatively good agreement: the interannual variability of SSS derived from the CROCO simulations (Figure 5a) correctly represents the main variability compared to CCI and GLORYS. There are differences between CROCO simulations, which can reach 0.2-0.3 pss during the rainy season at the end of the year (e.g., 2014, 2015). For the rest of the year, the differences remain negligible. The strongest CROCO SSS anomalies are generally

produced by CROCOglofas.

In 2011, 2015 and 2018 strong SSS anomalies were observed in the CCI and GLORYS products (see grey shading in Figure 5), and were well represented by the CROCO simulations (Fig.5a). Figure 6 shows that the spatial distributions of SSS simulated by CROCOimerg and those of CCI SSS over the region for these three years are very similar during these time periods, which are now studied in more detail.






**Figure 6: a, d, g): Anomalies of the terms of the salinity balance equation (in pss/day) for CROCOimerg. The yellow line represents the effect of the atmospheric forcing, the blue line the lumped advection term (horizontal and vertical, including runoff forcing), the pink line the entrainment term at the base of the mixed layer. The red line is the SSS rate term, which corresponds approximately to the sum of the other three terms (vertical and horizontal diffusion are negligible). The grey shading marks time periods of strong salinity variations. b, e, h) simulated SSS maps (in pss) averaged over 3 months for CROCOimerg. Arrows show the surface currents anomalies. c, f, i): CCI SSS maps (in pss) averaged over 3 months. Contours show IMERG precipitations (in mm/d, contour spacing is 1 mm/d). Different time periods are shown: 2011 (a, b, c), 2015 (d, e, f) and 2018 (g, h, i).**





### 3.3.1 Positive SSS anomaly in 2011

Around mid-2011, CROCO simulates a steep SSS increase and positive anomaly, which is in good agreement with the CCI anomaly (Figure 5a). The GLORYS anomaly displays the same variation albeit with a lesser magnitude. Although all CROCO simulations reproduce the SSS increase, the CROCOisba anomaly is the closest to the CCI anomaly.

As it is found in all CROCO simulations, this anomaly thus is independent of the differences between precipitation and runoff forcing anomalies (Figure 5b,c). The fact that the SSS increase is also found in the simulation with climatological precipitations (CROCOprclm), and that the IMERG and ERA5 rainfall anomalies are of opposite sign (Figure 5c), suggests that this anomaly does not result from a rainfall anomaly. Moreover, the difference in runoff generates a difference in SSS anomalies that is only of second order. Consequently, the SSS increase must arise mainly from the ocean circulation. The salinity balance (Figure

6a.) confirms that a positive anomaly of the advection term is the main cause of the SSS increase. A closer analysis of the advection anomaly indicates that it is mainly due to the anomaly of currents (V'<S>, see Appendix D) related to an increase of the southward wind-driven coastal current (coastal jet) through the climatological poleward gradient of salinity (Figure 6b.).

### 3.3.2 Negative SSS anomaly in 2015

Starting in mid-2015, most CROCO simulations show a significant freshening (from -0.4 pss for CROCOglofas to -0.2 pss for the other simulations) which is in good agreement with CCI and GLORYS (Figure 5a). In contrast, simulations forced by climatological precipitations (CROCOprclm) display no anomaly, while simulations forced by climatological (CROCOroclm) and ISBA (CROCOimerg) runoff display a freshening of 50% weaker than with GLOFAS runoff (CROCOglofas) (Figure 5b).

This suggests that this freshening is initially due to the precipitation anomaly, followed by the subsequent runoff anomaly (Figure 5c). The analysis of the salinity balance equation confirms this hypothesis (Figure 6d.e.): the freshening is due firstly to the rain intensification (October-December 2015). It is then slightly reinforced by the runoff in December-January and by an anomalous salty water outflow at the northern boundary of the e-SNTA region (Appendix D). This anomalous advection of fresh water is amplified by the higher GLOFAS runoff in CROCOglofas. This effect is discussed in more details in sect. 4.


### 3.3.3 Positive SSS anomaly in 2018

In mid-2018, the SSS anomalies reach ~0.3 pss (Figure 5a). As in 2011, all CROCO simulations, including those with climatological forcing, reproduce this positive anomaly, which thus cannot be attributed to one particular forcing anomaly. Analysis of the salinity balance (Figure 6g) shows that the salty anomaly initially results from the strong negative precipitation

anomaly (i.e., rain deficit) found in both ERA5 and IMERG products (Figure 5c). The greater impact of the IMERG



precipitation anomaly on CROCOimerg SSS could be due to a more localized and intense precipitation anomaly in IMERG than in ERA5 (see Appendix A). This precipitation anomaly is accompanied by a large ISBA runoff negative anomaly (Figure 5c), which explains why CROCOprclm also simulates the positive SSS anomaly (albeit weaker than in CROCOimerg) without a precipitation anomaly, and why the simulations using ISBA runoff (CROCOimerg and CROCOisba) display a stronger

anomaly than that using GLOFAS runoff (CROCOglofas). Note the surprisingly large GloFAS runoff (Figure 5c), opposed to the rain deficit over the oceanic region during this period.

## 3.4 SSS Sensitivity to the freshwater forcings

In this section, we investigate more thoroughly the sensitivity of the simulated SSS to a modulation of runoff forcing, all other
model forcings being kept identical. We analyze the temporal variability of simulated salinity over the whole simulated period. Three test cases are set up using the simulations described in Table 1: the differences between simulations with a climatological runoff and a synoptic runoff (the GloFAS product) are first analyzed with regards to the interannual variability of forcing; then the difference in SSS induced by the use of the ISBA or GloFAS synoptic runoff products is analyzed. Last, the effects of SSS modulation by precipitation changes is presented.

The panels in Figure 7 depict these case studies. For each case, the difference in SSS between the studied simulations is shown in red, the differences in forcings in the e-SNTA region in solid brown lines, and those occurring south of the e-SNTA region ([5°N 10°N - 10°W 20°W]) in dashed lines. The region south of the e-SNTA is indeed the site of strong freshwater influx, and the general oceanic circulation tends to advect these waters northward into the e-SNTA region. Correlation ($r^2$) is calculated while allowing for a temporal lag, over the e-SNTA region and the region south of the e-SNTA.








**Figure 7: a) GloFAS runoff anomalies (blue) and difference between SSS anomalies (red) (CROCOglofas-CROCOroclm). b) Differences between GloFAS and ISBA runoffs anomalies (blue) and differences between SSS anomalies (red) CROCOglofas-CROCOisba). Note that CROCOglofas, CROCOroclm CROCOisba are forced by the same ERA5 precipitation fields. In a) and b),**





**the sum of the runoff anomalies of rivers flowing directly into the e-SNTA (solid line), and the sum of the runoff anomalies of rivers flowing south of the e-SNTA (dashed line) (Figure 1). c) Differences between IMERG and ERA5 precipitation anomalies (in m$^3$/d, blue line) in e-SNTA (solid line) and south of e-SNTA (dashed line), and differences between SSS anomalies in e-SNTA (in pss, red line) from simulations using IMERG (CROCOimerg) and ERA5 (CROCOisba) with the same ISBA runoff forcing. To ease the reading of the figure, the sign of precipitation anomalies has been reversed. Correlation (r$^2$) and time lag between the time series of SSS anomaly and of the sum of the freshwater flux in and south of the e-SNTA regions are shown in the top center of the panel.**

### 3.4.1 SSS sensitivity to interannual versus climatological runoff

The influence of GloFAS interannual runoff variability on salinity is investigated by comparing the CROCOroclm SSS to the CROCOglofas SSS. The other forcings (*i.e.,* ERA5) are kept identical between the two simulations (Table 1) so that the differences observed in SSS is mainly the consequence of the difference in river outflow forcing and its effect on nearshore ocean dynamics. Note that SSS differences can also arise from differences in mesoscale circulation due to dynamical (chaotic) nonlinearities or intrinsic variability unrelated to the forcings.

Interannual runoff variability has a significant effect on SSS (Figure 7a). The GloFAS runoff interannual variability is indeed correlated with the difference between CROCOglofas SSS and CROCOroclm SSS (r$^2$ of 0.38 for a lag of 54 days; Figure 7a). Differences in SSS may be due to both a local and a remote runoff anomaly (i.e., south of the e-SNTA) (Figure 7a, dotted line).

The salinity balance for CROCOglofas and CROCOroclm (not shown) indicates that, as for the climatological cycle (Appendix B), a difference in runoff is partly compensated by a difference in entrainment of opposite sign. There is indeed a correlation of 0.98 between the runoff interannual anomaly and the entrainment terms difference, 75% of the runoffs anomalies being compensated by entrainment on average. The lag time (54 days) between runoffs anomalies and SSS difference is likely due to a localized (nearshore) effect of the runoffs taking time to spread offshore and to modify the ocean surface layer over the entire e-SNTA.

### 3.4.2 SSS sensitivity to a change in runoff interannual variability.

The difference between ISBA and GloFAS runoff anomalies is shown on Figure 7b for rivers flowing directly into the e-SNTA (local runoff, solid line) and for rivers flowing in the south of the e-SNTA (dotted lines) (Figure 1). The differences between ISBA and GLOFAS runoffs are greater than the anomalies of either runoff taken independently, as the anomalies are frequently of opposite sign (Figure 5c.). As a consequence, the impact of the total runoff difference on salinity is larger than in the previous case study, with differences reaching 0.3 pss. A r$^2$ of 0.56 is obtained between the difference in runoffs and the difference in SSS, with a time lag of 68 days; difference of runoffs is also offset by the entrainment at the bottom of the mixed



layer: runoff difference and entrainment difference have indeed a correlation of 0.98 and entrainment compensates on average for 84 % of the runoff difference, with a time lag of 13 days.

In these two case studies, the effect of runoff is variable: small deviations from the climatology can generate significant differences in SSS as in 2015 (Figure 7a,b), while large runoff anomalies sometimes have a limited effect as in 2017 or 2018. These differences in behavior can be explained by surface current anomalies: in 2015, a northwesterly current transports the SSS anomaly linked to river flow, so that it has a greater impact on the mean SSS of the e-SNTA region (Figure 6e). Thus, averaged SSS is particularly affected by a small change in runoff. Conversely, in 2018, (Figure 5.c) the large difference ($1.2.10^9$

$m^3d^{-1}$) between the two forcings (Figure 7b), with anomalies of opposite signs (Figure 5c), coincided with a northerly wind anomaly (not shown). The SSS anomaly produced by runoff in 2018 is therefore confined to the coasts south of the e-SNTA, it does not spread northward, and it has a relatively little impact (0.22 pss) on the mean salinity of the area (Figure 6h.).

### 3.4.3 SSS sensitivity to a change in rain rate interannual variability


The effect of a change in precipitation on the modeled salinity, more specifically on the difference induced by ERA5 (CROCOisba) and IMERG (CROCOimerg) synoptic precipitation products, is shown on Figure 7c.

The differences between the IMERG and ERA5 mean precipitation fields are small in comparison with the amplitude of their climatologies, (Figure 5c, Appendix B), but the aggregated freshwater fluxes over the e-SNTA are of the same order as the

differences between runoffs forcings. The differences in SSS resulting from this difference in rain rate (Figure 7c) are weaker than those associated with runoff differences and not strongly driven by the effect of rain, in contrast to the effect of river discharge (sect. 3.4), as shown by the low correlation ($r^2$=0.19). Note, however, that differences in SSS seem to be linked to differences in precipitation over certain periods (e.g., mid 2011, mid-2013, late 2014). As for runoff, the salinity balance of CROCOimerg shows that an anomaly in the rain rate forcing term is nearly totally compensated by an anomaly in the

entrainment term (Figure 6a.d.g.). This adjustment is exactly correlated with the rainfall difference ($r^2$=0.99), and the entrainment difference compensates in average 98 % of the forcing term difference. The weaker correlation with precipitation than with runoff anomalies could be explained by the fact that precipitation anomalies are weaker locally but spread over a larger region than runoffs.





## 4 Conclusions and discussion

**4.1 Conclusions**

SSS are simulated over the period 2010-2019 using an ocean circulation regional model (CROCO) off the west African region forced by various precipitation and river runoff products. The simulated SSS are compared to various local and global data sets: the CCI satellite product, the GLORYS reanalysis, the ARGO floats data base, a coastal mooring and TSG measurements. The comparisons show that modelled SSS is systematically too high north of 15°N. Nevertheless, comparison with a coastal

mooring 20 km off the coasts of southern Senegal show an excellent agreement. The simulation forced by IMERG precipitation and ISBA runoff is slightly closer to the observations than the other simulations, on average over the period of study.

The simulated SSS are analysed in terms of seasonal cycle and interannual anomalies averaged over the eastern Southern North Tropical Atlantic (e-SNTA). At first order, the amplitude and phase of the SSS seasonal cycle are only slightly modified by the different precipitation and runoff products used as model forcing. However, there is a time lag of about 2 weeks between

the simulations, which corresponds to a shift in the seasonal cycles of the two runoff products, and a difference in amplitude of 0.1 pss, which may be due to a difference in precipitation forcing. The seasonal cycles of the CCI satellite data and the GLORYS reanalysis are also out of phase by about 2 weeks, which could originate from the climatological runoffs used in the GLORYS reanalysis. Analysis of the modelled mixed layer SSS budget indicates that the SSS decrease during the rainy season is driven initially by precipitation and a few weeks later by river runoff by means of horizontal advection. Note that these

negative trends are partly (nearly fully) compensated for runoffs (precipitation) by entrainment of relatively saline subsurface water into the mixed layer.

Despite the systematic model bias, modelled and observed SSS interannual variations are very consistent. Large SSS anomalies are often correlated with large precipitation or runoffs anomalies within the e-SNTA and from neighbouring regions whose surface waters are then advected by surface currents into the e-SNTA. However, a propagation of the river plume is not

systematic, and depends in particular on the wind-driven surface circulation patterns.

A study of the sensitivity of SSS to precipitation and runoff shows a different response of the surface ocean to the two types of forcing. A difference in precipitation is almost totally compensated by entrainment, while a difference in runoff is compensated by between 75% and 84% on average. For a change in forcing of an equivalent order in terms of mean freshwater input, surface salinity is therefore more impacted by river runoffs than by precipitation.






## 4.2 Disparity of river discharge products

This study also highlights the disparity between two river discharge products available for West Africa. The GloFAS and the ISBA products are rather consistent in terms of seasonal cycle amplitude, with a maximum difference of $3.10^8$ m³/day (Appendix B), but the phasing of the cycle is shifted by about 2 weeks and interannual anomalies are frequently of opposite

555        sign (Figure 5c). This is in line with results from (Decharme et al. 2019) showing that runoffs simulated by different hydrological models driven by various precipitation products have a significant disparity in the intensities and phase of seasonal runoff cycles (*e.g.*, see Fig 14 of (Decharme et al. 2019)).

To understand the origin of these disparities, the runoff anomalies of each product are compared with the rainfall anomalies over the catchment areas of the rivers flowing through the study region (see Figure 1 for catchment delimitation). Runoff

560        anomalies of each product are strongly correlated with the rainfall anomalies used in the hydrological models (Figure 8): GloFAS runoff anomalies are correlated with an r² of 0.87 to the ERA5 rainfall anomalies over the catchment area, with a time lag of 22 days, and ISBA runoffs anomalies are correlated with the IMERG rainfall anomalies over the catchment area with an r² of 0.56, and with a time lag of 33 days. This suggests that the quality of runoffs estimation is highly dependent on the quality of the estimation of rainfall on land. Comparisons of modelled SSS with *in situ* SSS over the

565        entire period (2010-2019) show slightly better results in simulations forced by ISBA, but using the GloFAS (resp. ISBA) product leads to more accurate modelled SSS at the beginning (resp. end) of the studied period. In conclusion, the present study presents an indirect evaluation of runoff interannual variability by analyzing their impact on surface ocean salinity.

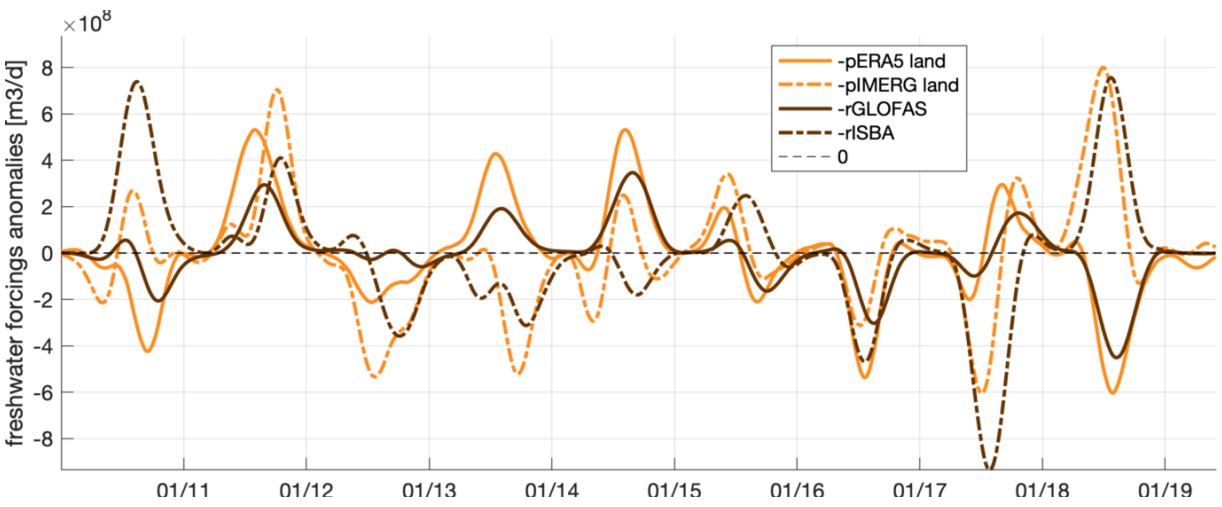


**Figure 8: Runoff anomalies (brown) and watershed precipitation anomalies (orange), for GloFAS and ERA5 (solid lines) and ISBA and IMERG (dashed lines), in m³/d, over the catchment areas of the rivers flowing through the study region (see Figure 1 for catchment delimitation).**



### 4.3 CROCO SSS uncertainties

Over several time periods (*e.g.*, 2010, 2013, 2014 and 2016; Figure 5a), the CROCO SSS anomalies are markedly different from those of the reference products (GLORYS and CCI). In 2010, the CROCO anomalies appear to be underestimated compared with CCI, which shows a very strong freshening (~-0.4pss) during most of the year (Figure 5a). However, this period corresponds to the beginning of the CCI product time series, during which the SMOS calibration was still unstable. The GLORYS product also shows a freshening (-0.15 pss), but weaker than in CCI and more similar to the CROCO SSS anomaly. The CROCOglofas simulation is the most accurate in 2010 with respect to CCI and GLORYS. In 2013, there is a large disparity in SSS (up to 0.3 pss) between the various CROCO simulations. The CROCOglofas SSS displays a very weak anomaly, similar to CCI, while all the other CROCO simulations show a moderate freshening (-0.15 pss to -0.28 pss). These discrepancies could be explained by the relatively large difference in runoff during this time period (Figure 5c). CROCOglofas is less biased than CROCOisba with respect to CCI and GLORYS, suggesting that the GloFAS 2013 anomalously low runoff (i.e., positive anomaly) is more realistic. In contrast, in 2014, CROCOglofas shows an unrealistic, high positive SSS anomaly, not found in the other simulations. This strong anomaly is associated with an anomalously low GloFAS runoff, in contrast to the ISBA anomalously high runoff driving the more realistic SSS anomaly in CROCOisba. In 2016, the CROCO simulations are in relatively good agreement with one another, but display a much stronger freshening than found in both reference products. Only CROCOprclm does simulate correctly the moderate freshening (Figure 5b, see magenta dotted line). Analysis of the salinity balance shows that advection drives the overly strong freshening (Appendix D), which may result from surface currents driven by unrealistic ERA5 winds in this time period. However, intrinsic variability may play a role as CROCOprclm is also driven by ERA5 winds.

### 4.4 Impact of a monthly climatological precipitation forcing

The CROCOprclm simulation was designed to suppress the effect of interannual precipitation variability on salinity. However, the calculation of a monthly climatological precipitation field by averaging monthly precipitation rates from various years drastically changes the distribution of precipitation (Appendix A.2) by smoothing and attenuating the highly localized precipitation phenomena. Climatological values do not exceed $2.10^{-2}$ m/day, which is too low for synoptic fields. So, rather than the effect of temporal precipitation anomalies, this simulation highlights the effect of a reduction of spatial rainfall variability. To reproduce a commonly observed forcing, it would be better to design a daily climatological field consisting of a succession of typical years with realistic precipitation distribution, but scaled to a climatology in terms of total precipitation quantities. Such experiments are planned for future studies.






## 4.5 Origin of CROCO SSS uncertainties

As seen in sect. 3.2.2, the CROCO SSS in all simulations were too high with respect to observations, mainly north of Cape Verde (15°N). This SSS bias is in fact associated with a positive temperature bias of ~0.5-1.5°C (Appendix E). Such an SST bias is estimated to lead to an excess of evaporation that could explain about one third of the SSS bias (see histograms in
appendix E). The remaining SSS biases could be due to a salinity bias in the subsurface waters transported to the surface layer by coastal upwelling and then offshore by Ekman currents.

Other processes neglected or misrepresented in the regional model may impact the SSS bias. First, the salinity of the river inflows (i.e., the runoff salinity), set to 15 pss in our study, may have an impact. This choice is debatable as it is expected that the salinity from rivers would be closer to 0 pss. For example, salinity gradually increases from ~0 pss at ~7 km from the coast
to 10 pss at the estuary mouth of the Suwannee River in West Florida (Laurel-Castillo and Valle-Levinson (2023)).

Most West African river mouths have the particularity of being located near very flat coasts, which promotes the formation of large estuaries, despite their relatively low runoff (Descroix et al. 2020). These large estuaries allow the intrusion of seawater inland and facilitate water evaporation.  Salinity in the Senegal River reach a minimum of 10 pss in October, at the peak of the flow, and 35 pss in winter (Mikhailov and Isupova (2008)). The Sine Saloum and Casamance rivers even have
inverse estuaries, with estuarine salinity higher than that of the ocean (Pagès and Citeau 1990, Descroix et al. 2020).

The value of 15 pss was chosen considering mixing between river waters and seawater as well as evaporation inside the estuary. However, the effect of a change in CROCO runoff salinity can impact on SSS in the freshwater plume: a sensitivity study shows that setting the runoff salinity to 1 pss instead of 15 pss can lead to a decrease in SSS of about 1 pss in regions traversed by the plume (see Appendix F). Future studies should take into account the seasonal variability of runoff salinity,
when data is available. It should be noted that this effect on SSS from a reduction in runoff salinity has little impact offshore of the Senegal river (Appendix F) and thus does not explain the positive SSS bias north of 15°N (Figure 4).

Last, tidal effects were not taken into account in our simulations, mainly to reduce computing time (a short model time step is required in the presence of strong tidal currents). However, it has been shown in the Amazon plume region that tides can impact plume propagation (Ruault et al. 2020) and conversely, that river flows can enhance tidal elevation (Durand
et al. 2022). A sensitivity study of SSS to tides shows that tides can, in some cases, cause an increase in average SSS in the e-SNTA region of about 1 pss due to increased vertical mixing (Appendix F). Thus, including this effect thus would not reduce the positive bias north of 15°N and the impact on our results is likely to be weak. However, a more detailed evaluation of the tidal effect on river plume propagation in our region of interest would be needed to confirm these results.



Data availability: TSG data are available at https://doi.org/10.6096/SSS-LEGOS; Argo data have been downloaded from Pi-
MEP database on https://pimep.ifremer.fr/diffusion/data/cci-l4-esa-merged-oi-v3.2-7dr/argo/. Melax data are available on
https://zenodo.org/records/4095436. CCI data are available at The Centre for Environmental Data Analysis (CEDA)
(https://dx.doi.org/10.5285/5920a2c77e3c45339477acd31ce62c3c). GLORYS reanalysis data are available on
https://doi.org/10.48670/moi-00021; the ERA5 dataset provided by the ECMWF is available on
https://cds.climate.copernicus.eu/cdsapp#!/dataset/reanalysis-era5-single-levels?tab= overview. CROCO model informations
are available on https://www.CROCO-ocean.org/ (Hilt et al. 2020). Given the large size of the modeling experiment outputs
(∼1.6 TB for each simulation), the dataset is not stored online and can be shared upon request to the corresponding authors.
The ISBA-CTRIP data are available from the authors (Decharme et al. 2019). IMERG data are available on
https://disc.gsfc.nasa.gov/datasets/GPM_3IMERGDF_07/summary?keywords=%22IMERG%20final%22; GloFAS data are
available on https://cds.climate.copernicus.eu/cdsapp#!/provider/provider-cems-without?tab=overview.

Author contributions: CTM conducted the study and wrote the initial version of the paper. CTM, VE, JB designed the
simulations, CTM prepared the forcing fields, VE performed the simulations and CTM analyzed them. CTM, JB, and JLV
performed the satellite data processing and analysis. AL collected the Melax in-situ data. CTM, JB, VE, AL participated in the
discussion of the results. All authors have read, improved, and agreed with the content of the paper.

Competing interests: The contact author has declared that none of the authors has any competing interests.

Acknowledgements: This work is part of the PhD of Clovis Thouvenin-Masson funded by CNES and ACRI-st. We also
received funding support from ESA CCI+SSS project (4000123663/18/I-NB) and CNES (the French National Center for Space
Studies) through the TOSCA SMOS-Ocean. CROCO modelling experiments were conducted on the IDRIS HPC Jean-Zay
under DARI projects A0110101140 and A0130101140. GLORYS global ocean model outputs were provided by the
Copernicus Marine Environment Monitoring Service (CMEMS). We thank X. Perrot for discussions about rain rates products,
M. Gevaudan, J. Jouanno and S. Biancamaria for providing precious information about ISBA-CTRIP.



## Appendix A: Distribution of precipitation forcings


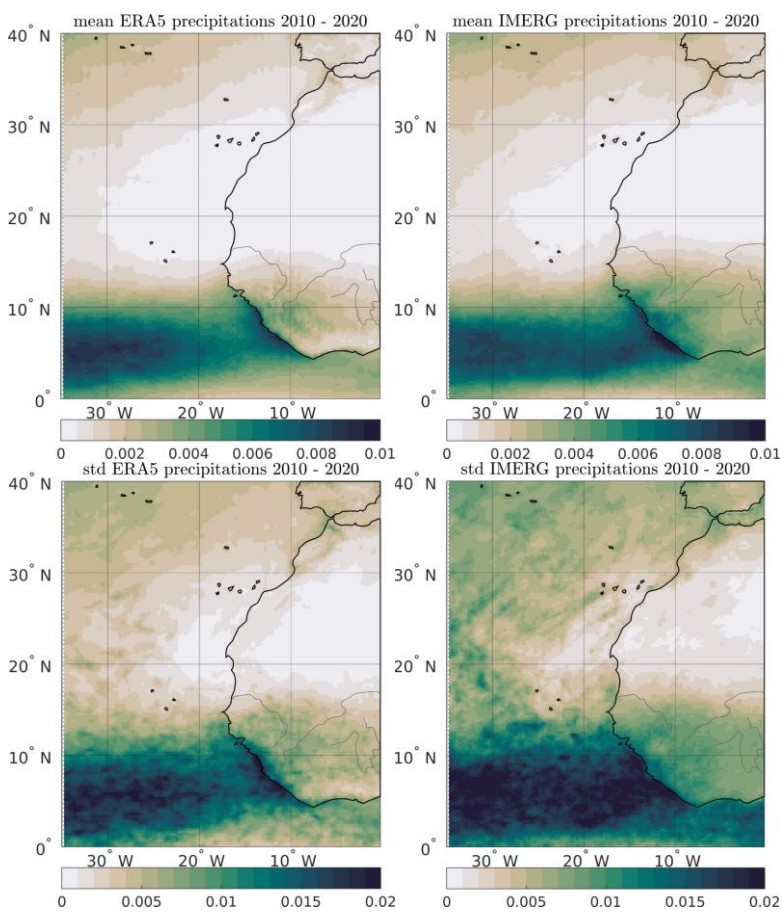

**Figure A.1: Distribution of ERA5 (left) and IMERG (right) precipitation (in m/day). Average precipitation between 2010 and 2019 (top), standard deviation of precipitation over the same period (bottom).**



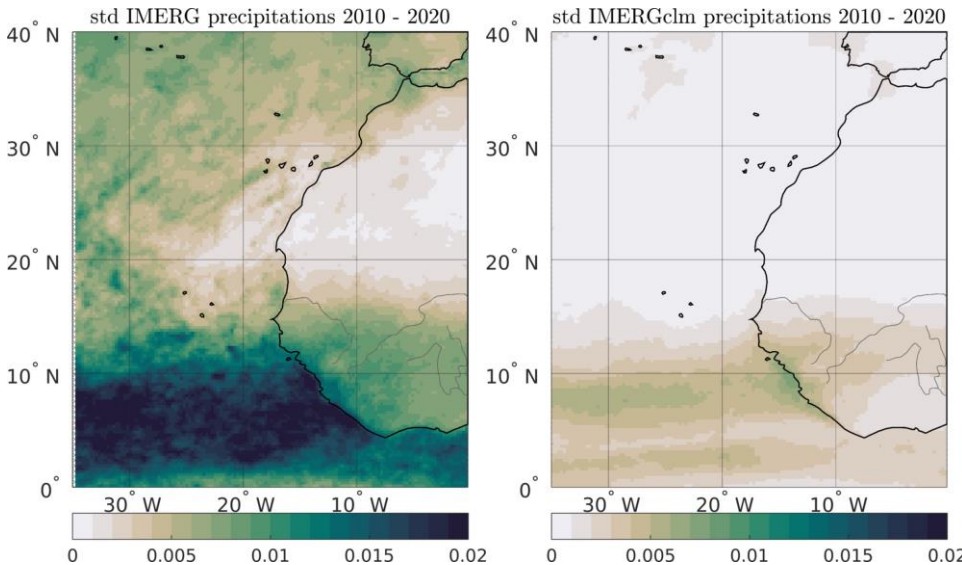

**Figure A.2: Standard deviation of IMERG precipitation (in m/day) (left) and the one used in the CROCOprclm simulation (right) over 2010-2020.**

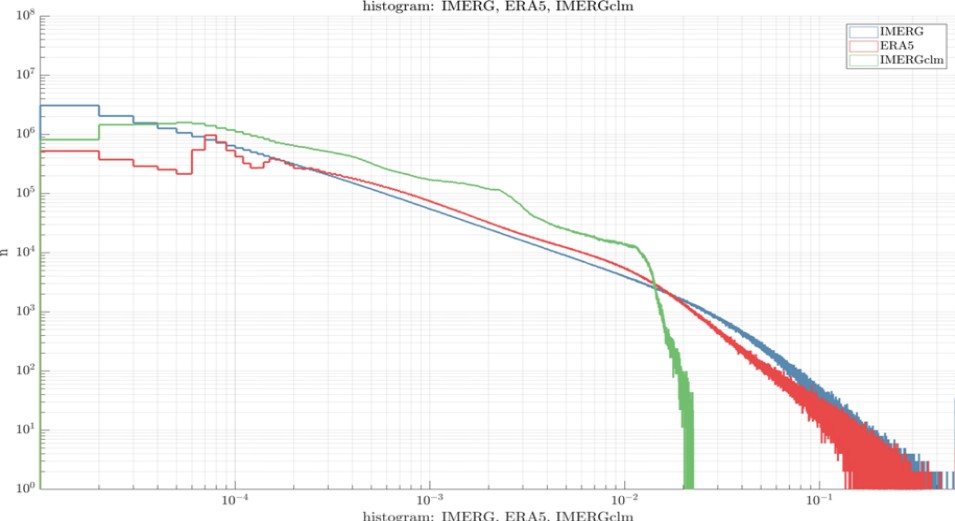

**Figure A.3: Histogram of the ERA5 precipitation (red), IMERG precipitation (blue) and monthly-mean climatological precipitation (in m/day) used in the CROCOprclm simulation (green). Note that scales are logarithmic.**





## Appendix B: Mean seasonal cycle in e-SNTA


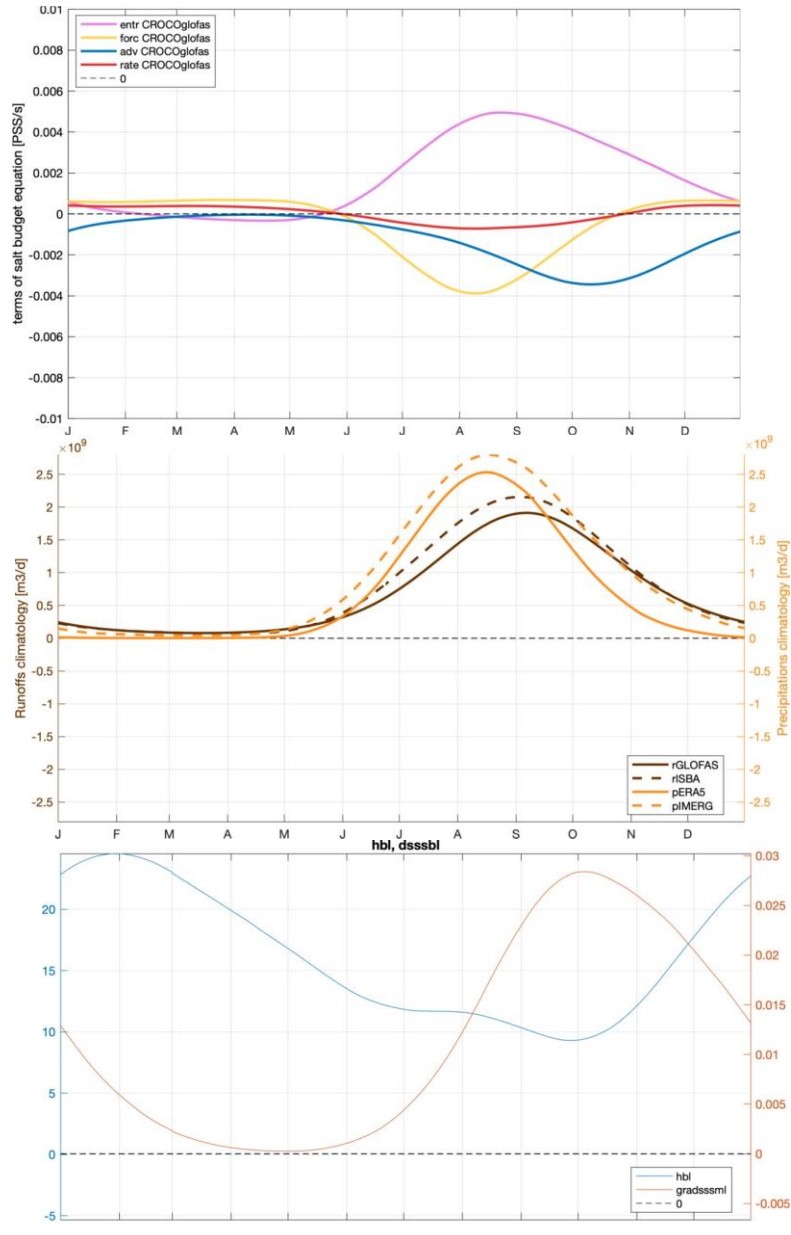

**Figure B.1: Seasonal cycle of the SSS budget in the e-SNTA region (simulation CROCOglofas). (top) Trends of the salinity balance equation (in pss/day). The orange line represents the effect of the atmospheric forcing, the yellow line the lumped advection term (horizontal and vertical, including runoff forcing), the pink line the entrainment term at the base of the mixed layer. The red line is**

**the SSS rate term, which corresponds approximately to the sum of the other three terms (vertical and horizontal diffusion are**



negligible). (middle) ERA5 and IMERG precipitation (orange, in m³/day), GloFAS and ISBA runoff (yellow) climatologies (in m³/day). (bottom) Mixed layer depth anomalies (in m; blue) and salinity gradient at the base of the mixed layer (in pss/m; red).

## Appendix C: salinity balance at Melax



**Figure C.1: (a)** Anomalies in the terms (in pss/day) of the salinity balance equation of the CROCOglofas simulation and **(b)** anomalies in the corresponding forcings at the Melax point. The colors in (a) are identical to those in Fig. B1a. The colors in (b) are identical to those in Fig. B1b.







## Appendix D: Salt flux at the border of e-SNTA for CROCOimerg simulation

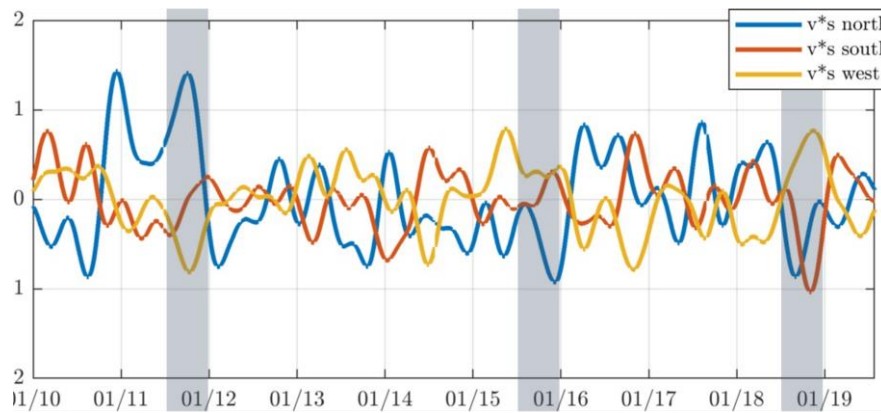


**Figure D.1: Band-passed anomalies (using a three-month running mean) in anomalous salt flux (V'.<S>) through the boundaries of the e-SNTA region due to a surface current anomaly (V') multiplied by climatological salinity <S> at the border. Blue line: flux through the northern boundary; red line: flux through the southern boundary; yellow line: flux through the western boundary. Positive flux indicates a salt input in the e-SNTA region.**




**Appendix E: SST and SSS biases in CROCO simulations**


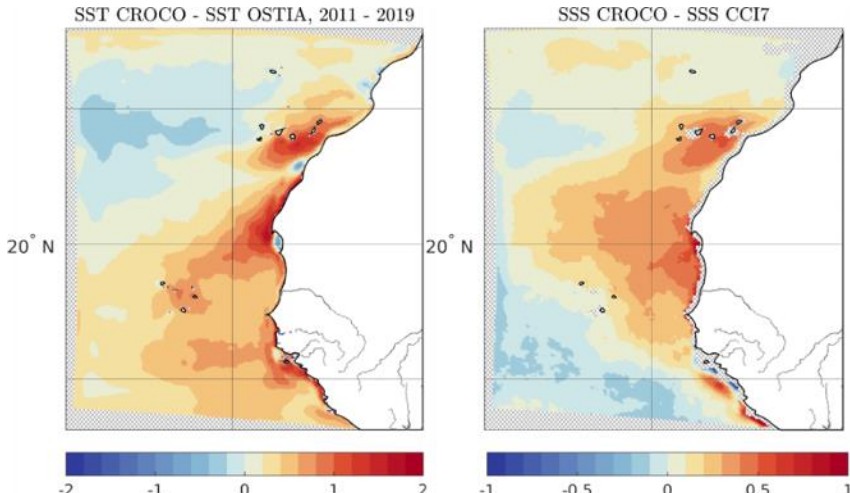

**Figure E.1: (left) Average SST bias (in °C) between CROCOGloFAS simulated SST and OSTIA SST and (right) average SSS bias (in pss) between the CROCOGloFAS simulated SSS and CCI SSS .**


We observe a bias of the model in SST compared with the OSTIA (Operational Sea Surface Temperature and Ice Analysis; Donlon et al. (2012)) satellite SST (Figure E.1a). Here we seek to determine whether this SST bias may be at the origin of the SSS bias discussed in the body of the article (Figure E.1b). Indeed, a bias in SST generates an evaporation deviation, which has an influence on salinity in the mixed layer. The SSS bias resulting from the SST bias is estimated to first order using the

bulk heat flux equations by the method described below: it is of the same order of magnitude as the SSS biases observed between CROCO and CCI (Figure E.1).

Bulk formulations describe flows at the air-sea interface, including evaporation:

$$E = \rho C_q (q_0 - q_z) u$$

With E the evaporation, $\rho$ the density, $C_q$ the Bulk coefficient, u the wind speed at altitude z above the surface. $q_z$ represents the specific humidity at altitude z, and $q_0$ the saturation specific humidity of the air at surface temperature $\theta_0$;
$q_z$ and $q_0$ are dependent on surface temperature and pressure, and are written as:

$$q_z = q_{sat}(d_z, SLP)$$





$$q_0 = 0.98 q_{sat}(\theta_0, SLP)$$

Noting SLP as the air pressure at the surface and $e_{sat}$ as the partial pressure of water vapor, we can calculate the specific humidity at a temperature T from the following formula:

$$q_{sat}(T, SLP) = \frac{\epsilon e_{sat}(T)}{SLP - (1-\epsilon)e_{sat}(T)} \quad \epsilon = 0.62$$


$e_{sat}$ can be determined by the Goff Gratch equation:

$$log_{10}(e_{sat}(T)) = 10.79574(1 - T_0/T)$$
$$-5.028 log_{10}(T/T_0)$$
$$T_0 = 273.16$$
$$+1.5047510^{-4}[1 - 10^{-82969(T/T_0-1)}]$$
$$+0.4287310^{-3}[10^{4.76955(1-T_0/T)} - 1]$$
$$+0.78614$$

Noting SSTi as the model SST and SSTf as the SST Ostia, switching from one to the other generates a difference in evaporation $\Delta E$ :

$$\frac{\Delta E}{E_i} = \frac{q_{0f} - q_{0i}}{q_{0i}} = \frac{q_{SAT}(T_f, SLP) - q_{SAT}(T_i, SLP)}{q_{SAT}(T_i, SLP)}$$


This difference is then incorporated into the salinity balance equation (equation 6), with the other terms (including mixed layer depth) unchanged at first order:

$$\Delta \, \partial_t S = - \frac{\Delta E}{E_i} . E_i . \frac{S}{H}\Big|$$
.

This gives a change in the SSS rate at each time step, which can be added to the model's salinity trend. The mean deviation obtained is of the same order of magnitude as the deviation between CROCO SSS and CCI SSS, although generally smaller.



The geographical pattern (not shown) is similar, although this is a rough estimate that does not take into account, for example,

the advection of saltier waters obtained after this correction.

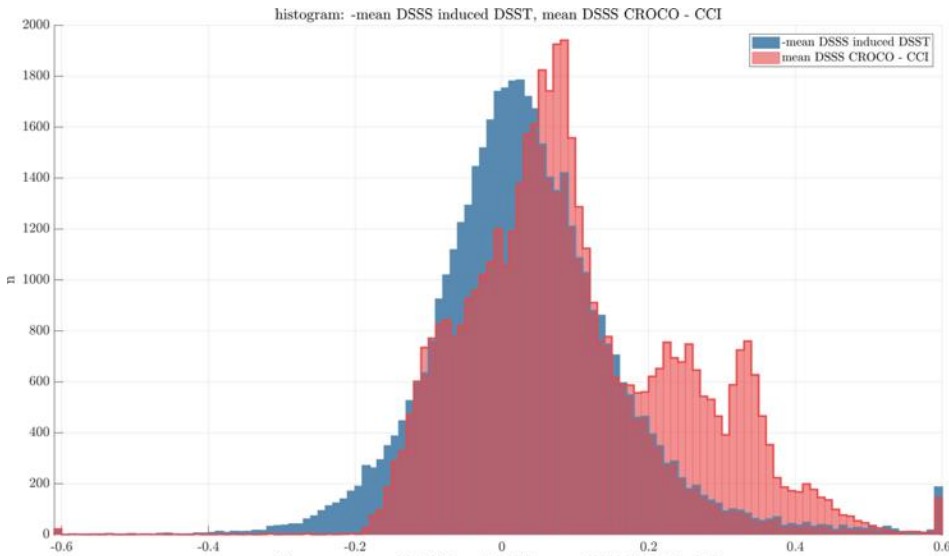

**Figure E.2: histogram showing the salinity bias between the CROCOglofas simulation and CCI (red) and the expected SSS bias**

**linked to the SST bias (blue), in pss.**





**Appendix F: Sensitivity of SSS to river water salinity and tides.**

**Figure F.1: Effect of tides on Sea Surface Salinity (SSS), in pss, in the e-SNTA region in 2015. The maps show the simulated salinity**
**with tides (a, c, e, g) and the difference from simulations obtained without tides (b, d, f, h). Maps averaged over 3 months: JFM**
**(January to March) (a, b), AMJ (April to June) (c, d), JAS (July to September) (e, f), OND (October to December) (g, h). The tidal**
**effect on SSS, leading to an increase of SSS in the plume, is mostly seen in OND (h).**





**Figure F.2: Effect of a change in river salinity on Sea Surface Salinity (SSS), in practical salinity units (psu), in the e-SNTA region in 2015. The maps initially show the simulated salinity with river salinity of 1 pss (a, c, e, g, i, k) and the difference from simulations obtained with river salinity of 15 pss (b, d, f, h, j, l). Maps averaged over 1 month: July (a, b), August (c, d), September (e, f), October (g, h), November (i, j), December (k, l).**



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
