# Peer review of "Influence of river runoffs and precipitation on the seasonal and interannual variability of Sea Surface Salinity in East Northern Tropical Atlantic"

_EGUsphere, 2024_

## Author Comment (AC1)

Reviewer 2

This study uses a combination of model data along with reanalysis, *in situ* and satellite observations to understand the seasonal and interannual variability of sea surface salinity (SSS) along the Senegalese coast in the northeastern tropical Atlantic Ocean. Sensitivity runs from CROCO model forced with different precipitation and river runoff datasets are analyzed to infer the impact of different model forcings on the seasonal and interannual variations of SSS in the region. A detailed description of the model data validation against the *in situ*, satellite and reanalysis SSS is provided. The study finds that the modelled interannual SSS variability off the Senegalese coast is more sensitive to river runoff forcing rather than precipitation. The seasonal cycle in SSS however remains unaffected by the different model forcings of precipitation and river runoff.

The manuscript is generally well written with decent quality figures. However, the manuscript needs some re-organization with more clear captions for the figures including the ones in the Appendices. The novelty of this study lies in exploring the impact of different model forcings on the e-NTA coastal SSS rather than analysis of the processes contributing to the SSS variability. This needs to be highlighted and stated in the Introduction section clearly. The manuscript may be considered for publication after the authors have addressed the major and minor comments listed below.

Major comments:

1. Motivation and the main objectives of the study need to be clearly stated towards the end of Introduction section. The focus of the study is on understanding the impact of different types of river runoff and precipitation model forcings on the seasonal and interannual variability of coastal SSS in e-NTA ocean. The discussion related to processes impacting the SSS variability on seasonal and interannual timescales using salt balance seems very descriptive and lacks physical understanding of the processes.

**This point was raised by all reviewers, and it indeed seems necessary to clearly specify in the introduction the purpose of the paper, which is to understand the relative effect of different freshwater fluxes on the seasonal and interannual variability of salinity, using a case study in the e-NTA region. The introduction has been modified accordingly (lines 69 - 74):**
**"The aforementioned studies demonstrate the usefulness of salinity as a tracer for variations in the water cycle, both from the perspective of the seasonal cycle and interannual variability. However, to our knowledge, no study has yet focused on interannual variability. This is the aim of this study, in which we aim to differentiate the effects of precipitation from those of river discharges on coastal salinity in the e-NTA region. To achieve this, the surface ocean dynamics is simulated by the Coastal and Regional Ocean Community (CROCO) model with various configurations of climatological or interannual forcings."**

2. There are too many subsections which can be merged (especially in sections 2 and 4). The discussion related to salt balance figures in the appendix is vague and not easy to understand. It was really difficult to go back and forth from the appendix to main article while reading the salt balance part. I suggest moving the salt balance figures in appendices B and C to the main article and the model validation plots to the appendix.

**Following the reviewer's suggestion, we have reorganized the structure of these sections to simplify it and we have merged several subsections. Sections 2.2.4, 2.2.5, 2.2.6, 2.3, 2.4 are now merged in section 2.2. Sections 2.5 and 2.6 are now in section 2.3. Sections 4.2 and 4.4 are merged in section 4.2.**
**As suggested by the reviewer, we have moved the figures from Appendix B into the main body of the paper, incorporating them into Figure 2. However, we have kept Appendix C as it is cited only once in the paper, and we do not consider it an essential figure. We consider that model validation is an important part of the paper and given that the number of figures is not excessive, we have decided to keep the corresponding figures in the main body of the paper.**

3. All figures' captions need to be written more clearly. The labels are not captioned in a chronological order. For example, Figure 6 caption includes text related to panels (a, d, g), (b,e,h), (c,f,i). It was difficult to navigate through the panels while reading the caption. Also add product name and variable as text inside each panel ('CROCO SSS' or 'CCI SSS') to make it easy for reader to understand what is plotted. This applies for other figures as well.

We added titles on top of the three columns of Figure 6, and modified and clarified the legend of Figure 6 to ensure that the panels are in alphabetical order (lines 405 – 412):

"Figure 6: SSS anomalies over the three analysed periods: late 2011 (a, b, c), late 2015 (d, e, f), and late 2018 (g, h, i). Left column (a,d,g): anomalies of the terms in the salinity balance equation (in pss/day) for CROCOimerg. The color code used is the same as that used in Figure 2a. Only pixels where CCI data are available have been considered in generating these curves. The black dotted line is the ISBA runoffs anomaly (the y-axis has been reversed). The grey shading indicates time periods of strong salinity variations. Central column (b,e,h): simulated SSS maps (in pss) averaged over 3 months for CROCOimerg. Arrows show the surface currents anomalies. Right column (c, f, i): CCI SSS maps (in pss) averaged over 3 months. Grey contours depict IMERG precipitations (in mm/d, contour spacing is 1 mm/d; darker grey correspond to higher precipitation)."

All the legends of the article have been slightly modified for clarification.

4.  The discussion related to salt balances in Figure 6 for the 2011, 2015 and 2018 episodes is not clear. In 2011, the positive SSS anomaly is attributed to advection, but the forcing term also shows the same sign and has magnitude comparable to the advection term (Fig. 6a). For 2018 positive SSS anomaly case, the rate term is negative (Fig. 6g). This needs to be checked. The negative entrainment (or residual) term in Fig.6a,g doesn't make physical sense as you would expect SSS to increase if there is entrainment of deeper saltier water to the surface.

Regarding the year 2011, Reviewer 2 notes that the forcing term is of the same magnitude as the advection term and questions why the anomaly is attributed solely to advection. Here, it is the comparison of different simulations forced with various precipitation products (Figure 5b) that allows us to reach this conclusion: We find that the positive forcing term is systematically overcompensated by the entrainment term, thus it has only a minor impact on salinity. We have added a sentence to clarify this in the document (lines 419 - 421): "The fact that the SSS increase is also found in the simulation with climatological precipitations (CROCOprclm), and that the IMERG and ERA5 rainfall anomalies are of opposite sign (Figure 5c), suggests that this anomaly does not result from a rainfall anomaly."

In Figure 6, it is important to keep in mind that the time series represent the interannual anomalies of the terms in the salinity budget equation, not the terms themselves. Thus, the negative value of entrainment anomaly in 2011 and 2018 indicates that entrainment tends to add less salt to the mixed layer during these years compared to the climatology. Indeed, these anomalies are on the order of $-1.5 \times 10^{-3}$ pss/s, while the climatological term (see new Fig. 2c) is $\sim 5 \times 10^{-3}$ pss/s, meaning the total entrainment remains positive despite this negative anomaly.

The negative rate obtained in 2018 was indeed an inconsistency as presented in the first version of the paper. We thank the reviewer for pointing this inconsistency to us. The sign difference in the rate between Figure 5 and Figure 6 came from a difference in the flags applied: In Figure 5, we removed points too close to the coast that cannot be observed by satellites and thus are not represented in the CCI fields, ensuring a valid comparison between CCI and model simulations. In the previous version of the manuscript, we did not remove these points in Figure 6. By removing the same points and focusing exactly on the same region, we obtain the new Figure 6g (see below), which is consistent with the 2018 positive anomaly shown in Figure 5b. We have specified this application of flags in the legends of Figure 5 and Figure 6 (see previous comment).

[Figure]

5.  In Fig. 5, the 2010 negative SSS anomaly event is interesting. This event could also be analyzed in addition to the 2011, 2015 and 2018 events, if that's easy. Also, can you comment on why the freshwater forcing terms estimated from GLOFAS and ISBA have huge differences in 2010, 2017 and 2018 (Fig. 5c)?

**The 2010 negative anomaly also caught our attention. However, we chose not to focus on this year due to the disagreement between CCI, GLORYS and simulated SSS anomalies. The simulated anomalies are relatively weak while, during that period the CCI dataset relies only on SMOS dataset which absolute calibration in 2010 is questionable. A study of the terms in the salinity budget equation (not shown) shows that the 2010 anomaly is primarily due to a positive precipitation anomaly (i.e. deficit of precipitation), modulated by a runoff anomaly, which differs drastically between the forcing products (see Figure 5c). the year 2010 is discussed in section 4.1. (lines 568 - 572).**
**The large difference between the various runoff forcing products is indeed an important point to note. An explanation is provided in the discussion (section 4.2, lines 586 - 601), where we highlight the link between the runoffs (GLOFAS and ISBA) which are outputs of hydrological models and the precipitation products (ERA5 and IMERG) used to force the hydrological models. The significant disparity in river discharge estimates thus likely stems from disparities in precipitation estimates over the African continent between ERA5 and IMERG. We have added a sentence in the description of Figure 5 that refers to this discussion (lines 401 - 402): "There is a significant disparity in the anomalies of the two river discharge forcing products, with anomalies sometimes having opposite signs (Figure 5c). This disparity is explored in the discussion (Section 4.2)."**

6.  Figure 7 needs modification. There are no brown lines plotted in the figure (Line 456). For each case study, can you add spatial plots of SSS, currents with box regions marked for e-NTA and south of e-NTA? Select the period during which you observe the advection of freshwater from the southern region to the north. How is the lag determined? Is the correlation coefficient maximum at this lag period? Include a plot of the correlation coefficient as function of lag in appendix if possible.

**The mention of brown lines on line 456 was an error. Adding a map for each case study does not seem necessary to us. Indeed, the figures represent runoff differences and salinity differences, which are difficult**

to visualize on a map. A contour showing the location of the southern e-NTA region has been added to Figure 1.

The time lag is determined to maximize the correlation between the curves. This is explained in section 2.3 (Analysis of cross-correlations, lines 255 - 259). A figure showing the correlation curves as a function of the lag has been added in Supplementary file (Figure S.4):

[Figure]

Minor comments:

1. The title needs to be modified to make it relevant to the main results presented in the study.

**We have changed the title according to Reviewer 1's suggestion: "Influence of River Runoff and Precipitation on the Seasonal and Interannual Variability of Sea Surface Salinity in the East Northern Tropical Atlantic."**

2. Eastern Southern North Tropical Atlantic (e-SNTA) is confusing. I suggest changing it to east northern tropical Atlantic (e-NTA).

**e-NTA is indeed clearer. We have modified the text accordingly.**

3. Mark the 2011, 2015 and 2018 periods in Figure 7 as well.

**We have added the shaded bands on Figure 7.**

4. Cite the appendix figure number instead of just saying Appendix in the main article. For example, Fig. D.1 instead of Appendix D in line 591. Same applies elsewhere.

**Agreed. We have transferred the figures from the appendices to a supplementary materials file, where they are listed from S1 to S10, as suggested by Reviewer 3. References to these figures now use this new notation.**

5. Font size of axes labels and legend in Fig. 6, Fig. B.1 needs to be increased.

**We have increased the font size on the cited figures.**

6. Remove the label for zero line in the legend of figures 5 and 6.

**We removed the corresponding labels.**

Line 13 – "relatively high cumulative river discharge" – relative to what?

**The sentence was modified. We suppressed the term "relatively" (line 13).**

Line 14 – precipitations – precipitation

**Agreed**

Line 28 – Forcing does not create mixed layer but impacts the mixed layer depth and dynamics.

**The sentence has been modified (line 27 - 28): « Air Sea forcings (e.g., wind) generate turbulence in the surface layer, leading to the formation of a surface mixed layer »**

Line 30 – What does flows exogenous to ocean mean?

**We suppressed this term and reformulated as follows (lines 30 - 31):**
**"This layer receives various freshwater flows, such as precipitation and river discharge. »**

---

## Author Comment (AC2)

Reviewer 1:

**General comments:**

This study investigates seasonal and interannual variations of sea surface salinity (SSS) in the North-Eastern Tropical Atlantic by means of several observational products and high-resolution regional model simulations forced by different runoff and precipitation data sets. It provides a thorough comparison between these different products and highlights how the choice in forcing data sets can impact the simulation of surface salinity in ocean models.

The manuscript is generally well written and provides a number of interesting insights into the factors impacting surface salinity in this region. I find, however, that it reads rather technical and that the value of the study lies mainly in exploring differences between various precipitation and runoff data sets and their impact on simulations of salinity while the dynamical understanding of the salinity variability remains rather vague.

**Specific comments:**

**Major:**

1. I am missing a bit more of a motivation that is then revisited in the conclusions. Why are interannual variations in salinity important?

**We have emphasized the importance of studying salinity and its variability at the end of the introduction drawing on past studies that show the link between salinity variability and the water cycle (lines 69 - 74): "The aforementioned studies demonstrate the usefulness of salinity as a tracer for variations in the water cycle, both from the perspective of the seasonal cycle and interannual variability. However, to our knowledge, no study has yet focused on interannual variability. This is the aim of this study, in which we aim to differentiate the effects of precipitation from those of river discharges on coastal salinity in the e-NTA region. To achieve this, the surface ocean dynamics is simulated by the Coastal and Regional Ocean Community (CROCO) model with various configurations of climatological or interannual forcings."**

2. I find it very hard to bring together the first part of the results section (3.1) with the corresponding figure in the Appendix. Instead of just referring to Appendix B, it would be helpful to refer to specific subplots and lines (something like "red line in Fig. B1(a)"). Also, the legend entries are hard to interpret and don't seem to always match the figure caption.

**Following the reviewer's comment, we have included the figures from Appendix B into the main body of the article to facilitate reading and have modified all the legends of the figures in the paper for clarification. Appendices are now moved to a supplementary material file, where figures are numerated from S1 to S10, as suggested by reviewer 3, and this notation is used for references in the article.**

3. While the case studies of years with strong robust SSS anomalies in section 3.3 is very interesting, the 2018 case remains rather inconclusive. It didn't become clear to me what actually caused this anomaly.

**For the year 2018, the analysis is more complex because the SSS positive anomaly results from a combined effect of precipitation and river runoff: it is initiated by positive precipitation anomalies (i.e. rain deficit) in both ERA5 and IMERG datasets and is then modulated by river runoff anomalies of different signs depending on the product used. We also corrected a small inconsistency in the computation of the trend anomalies, slightly modifying Figure 6g (see answer of Major comment 4 of reviewer 2). We have modified section 3.3.3 to clarify this point (see lines 451 - 456):**

**"In conclusion, the 2018 SSS anomaly is due to the combined effects of precipitation anomalies and river discharges. It is primarily caused by a strong precipitation negative anomaly (observed in both forcing datasets), which is not entirely compensated by entrainment (Figure 6c). This is then accompanied by a river discharge negative (positive) anomaly of ISBA (GloFAS) runoff, thereby accentuating (mitigating) the salinity anomaly through advection. This runoff anomaly explains the CROCOprclm SSS anomaly. The**

**large GloFAS runoff is surprising as it is opposite to the rain deficit over the oceanic region during this period (Figure 5c).”**

4. One of the main conclusions of the study is that in the SSS budgets, the precipitation term is largely compensated by the entrainment term. What drives this compensation, i.e. why does entrainment react to the surface freshwater input?

**Indeed, we found that the precipitation term is largely compensated by the entrainment term, both in the climatological cycle and during interannual anomalies. This can be explained as follows: when precipitation occurs, surface waters become less saline, and a vertical gradient of salinity is formed in the surface layer. As the mixed layer depth deepens during night-time, saline subsurface water is incorporated (entrained) into the mixed layers, leading to an increased mixed layer salinity. This diurnal salinisation of the mixed layer occurs even when the mixed layer tends to decrease at a seasonal time scale (Figure 2d). Thus, the larger the precipitation, the larger the salinity vertical gradient, the larger the entrainment and compensation by salinisation of the mixed layer.**

**We added a paragraph in the conclusion section 4 (lines 551 - 556)**

**Minor:**

1. I would suggest to reword the title to “Influence of Freshwater Fluxes on the Interannual Variability of Sea Surface Salinity in the North-Eastern Tropical Atlantic”

**We accept the title suggestion and have changed it accordingly.**

2. The region considered here can just be called “North-Eastern Tropical Atlantic” as in the title. There is no need to add an extra “southern” (line 13, 50 and elsewhere).

**Agreed**

3. As there are several units for salinity (psu, pss, g/kg) it would be good to comment on the unit used here.

**Agreed. We use the same unit (pss)**

4. Please specify the time period of all the used data sets.

**We have added information on the availability period of each dataset, as well as their resolution, where it was missing:**

**“TSG data are available from 1993 to present, between 5 to 15 m depth, and we use the hourly product.” (lines 125 – 126).**

**“ERA5 hourly fields are available over the period 1950-2023 at a horizontal resolution of 31 km” (lines 173 - 174)**

**“IMERG data are available from 2000 to present, at a resolution of 0.1° every half-hour.” (lines 185 - 186)**

**The GloFAS hydrological model simulations are available from 1979 to present at a daily and 0.1° resolution.” (line 197 - 198)**

**“ISBA-CTRIP data is available daily from 1979 to June 19, 2019, at a 0.5° resolution.” (line 209 - 210)**

5.  Section 2.2.4: I guess the model uses more than one baroclinic mode.

**There was a misunderstanding as CROCO does not compute the evolution of baroclinic modes separately. We rephrased as follows: "The slow mode and the fast barotropic mode are computed separately using a time-splitting algorithm (Shchepetkin and McWilliams 2009)" (Lines 142 – 144).**

6.  In line 301, it should probably read "deeper" instead of "thinner mixed layer"?

**Agreed.**

7.  In Figure 6, it would be helpful to also show SSS anomalies.

**After attempting to add salinity anomalies to Figure 6 (see below), we considered the resulting figure to be too cluttered. Moreover, the anomalies are already visible in Figure 5, and the salinity anomaly can be deduced from the "rate", which is its time derivative.**

[Figure]

8. Looking at Figure 5, I wouldn't say that that interannual variations are "very consistent" between model simulations and observations.

**We rephrased as follows (line 557): "Despite the systematic model bias, modelled and observed SSS interannual variations are overall in good agreement »**

9. There is a huge number of subsections, and I believe some of them could be merged. This applies to section 4.3 and 4.5. in particular.

**Sections 2.2.4, 2.2.5, 2.2.6, 2.3, 2.4 are now merged in section 2.2. Sections 2.5 and 2.6 are now in section 2.3. Sections 4.2 and 4.4 are merged in section 4.2.**

**Technical corrections:**

- line 28: As many waves are wind-forced themselves, waves shouldn't be lumped together with wind as a forcing.

**Agreed. We removed waves from the list. (Line 27)**

- line 30: Not sure what is meant by "exogenous" here.

**We suppressed this term.**

- line 31: "they lower the density" instead of "they make the density decrease"

**We reformulated the sentence as suggested.**

- line 139: "August" (with capital A)

**Agreed**.

- line 278: "linked to the salinity budget"

**Agreed.**

- line 298: I am not sure "attenuates" is a good expression in this context.

**We replace the corresponding sentence by: "The ocean transfers this freshwater input towards the ocean interior through vertical advection (not shown) and entrainment." (line 299)**

- line 377: "band-pass filtered" instead of "band-passed"

**Agreed**

- line 413: "lower magnitude"

**Agreed**

- line 456: There are no brown lines in Figure 7.

**The mention of brown lines on line 456 was an error.**

- line 553: It is not clear here whether "maximum difference" refers to the seasonal range or difference between products.

**We have specified to which quantity the difference applies (the seasonal range) (Line 591).**

---

## Author Comment (AC3)

Reviewer 3:

I agree with the two previous reviewers that this manuscript is more about runoff products validation. Because for interannual variability of mixed layer salinity in such a big region, the datasets (observations & model) cited in this manuscript are good enough. There's no need for such complex high-resolution sets of simulations. Therefore, the title, introduction, objectives, methodology needs to be reviewed substantially.

**Following reviewer 1 comment, we have modified the title of our study, which better reflects the aim of the present work. The introduction was also modified to clarify the purpose of the study (lines 69 - 74). However, the methodology was not modified, as we believe that our approach using a high-resolution model and a mixed layer salinity budget is sound. The high resolution of the model allows to represent the river inputs more precisely than a low-resolution model. It is possible that we would have obtained similar results using a model configuration at 0.25° like the one used by Camara et al. (2015). However, we have used a modelling tool recently developed by Ndoye et al. (2018) that is used in many studies by our group.**

The study region needs better reasoning. Currently the dashed black box in Figure 1 does not include the impacts of the whole merged catchment.

**The dashed box indicates the region of study for the ocean. Indeed, it does not include the rivers south of the box, but the low salinity waters produced by these runoffs are transported into the box by advection. Furthermore, we analyze the impact of the lumped river discharge south of the box in Figure 7 (see lines 494, 515).**

The datasets description lacks important specifications such as data period, temporal and spatial resolution.
**We have added information on the availability period of each dataset, as well as their resolution, where it was missing:**

**"TSG data are available from 1993 to present, between 5 to 15 m depth, and we use the hourly product." (lines 125 – 126).**

**"ERA5 hourly fields are available over the period 1950-2023 at a horizontal resolution of 31 km" (lines 173 - 174)**

**"IMERG data are available from 2000 to present, at a resolution of 0.1° every half-hour." (lines 185 - 186)**

**The GloFAS hydrological model simulations are available from 1979 to present at a daily and 0.1° resolution." (line 197 - 198)**

**"ISBA-CTRIP data is available daily from 1979 to June 19, 2019, at a 0.5° resolution." (line 209 - 210)**

The figures in the appendix should be included in a separated supplement document with increasing numbering order.

**Appendices are now moved to a supplementary material file, where figures are numerated from S1 to S10, as suggested.**

---

## Author Response (AR2)

Review for revised version of "Influence of river runoffs and precipitation on the seasonal and interannual variability of Sea Surface Salinity in East Northern Tropical Atlantic" by Thouvenin-Masson et al.

I appreciate the authors' effort in addressing the issues raised in my previous review. I find the revised manuscript to be improved and much easier to follow, but there are still a few issues that need to be taken care of before publication.

**We thank the reviewer for pointing out new inconsistencies and inaccuracies in the text. We have taken his comments into account as listed below. For reasons mentioned in remark D, we had to change the order of the figures (moving Figure 8 to the second position, to place it next to its first citation).**

A) While the motivation and goal of the study is given more clearly in the revised version, I think it could be highlighted even more in the abstract and introduction that the focus is on the impact of different precipitation and runoff forcing data sets on the simulation of salinity (rather than the dynamical understanding of salinity variability).

**We have revised the abstract and introduction to better highlight this aspect of the paper, as described below.**

**Abstract:**
**« The simulated SSS are compared with the Climate Change Initiative (CCI) satellite SSS, in situ SSS from Argo, ships and a coastal mooring, and the GLORYS reanalysis SSS. An analysis of the mixed layer salinity budget is then conducted. » (lines 17 - 19)**

**Instead of:**
**"The simulated SSS are compared with the Climate Change Initiative (CCI) SSS, in situ SSS from Argo, ships and a coastal mooring, and the GLORYS reanalysis SSS. The analysis of the salinity balance in the mixed layer is conducted to explore the dynamics influencing the SSS variability."**

**Introduction:**

**« These are the goals of this study, in which we aim (i) to differentiate the effects of precipitation from those of river discharges on coastal salinity in the e-NTA region, and (ii) to contrast the effects of different precipitation and runoff datasets on the simulated salinity." (lines 72 - 74)**

**Instead of:**
**"This is the aim of this study, in which we aim to differentiate the effects of precipitation from those of river discharges on coastal salinity in the e-NTA region."**

**« We estimate the seasonal cycle and interannual variation in salinity for each configuration, and intercompare these different configurations. Using a mixed-layer salinity balance, we identify the mechanisms through which river runoffs and precipitation alter the simulated SSS, employing a methodology similar to that of Camara et al. (2015). » (lines 77 - 79)**

**Instead of:**
**"We estimate the seasonal cycle and interannual variation in salinity for each configuration, intercompare these different configurations, and identify the physical drivers of SSS using a mixed-layer salinity balance, following a methodology similar to Camara et al. (2015)".**

B) I still find it hard to interpret the salt budget curves (Fig. 6) and bring them together with the corresponding text. Please make sure that it is always clear which line you are referring to.
As an example, the salinity anomaly is 2011 is attributed to advection but it looks like the advection term gets large only after the salinity increased.
Also, the negative precipitation anomaly in this case is argued to be compensated by entrainment. In 2018, the entrainment term is huge, much larger than just compensating the atmospheric forcing term, but there the

precipitation anomaly is argued to be important for the salinity anomaly. The advection term is much larger in 2018 than in 2011 but not mentioned to contribute at all.

**We have specified in the text which curves we are referring to, and we also refer to Fig 6**

**For the 2011 anomaly, there is indeed a cumulative effect of overcompensation of freshwater inputs by entrainment, followed by an advection anomaly, and we have clarified this in the text, as follows:**

**« The fact that the SSS increase also appears in the simulation with climatological precipitation (CROCOprclm), and that the IMERG and ERA5 precipitation anomalies are of opposite signs (Figure 6c), suggests that this anomaly does not result from a precipitation anomaly. Furthermore, the SSS anomaly is also present in the simulation with climatological runoff (CROCOroclm), and the changes in runoff forcing only generate second-order differences in the SSS anomalies (Figure 6b), indicating that the runoff anomaly is not the primary cause of the salinity anomaly. Consequently, the SSS increase must arise mainly from the ocean circulation. The salinity balance (Figure 7a) confirms that a positive anomaly in entrainment in summer (July-August), overcompensating a negative anomaly of atmospheric forcing, triggers the SSS increase, which is reinforced by a positive anomaly of advection in fall-early winter (September-January), in the case of CROCOimerg. » (lines 420 - 427)**

**Regarding the 2018 salinity anomaly, entrainment compensates for both the freshwater input from precipitation (forcing, yellow curve) and from rivers (included in advection, blue curve), leading to its high value. The discharge anomaly plays a significant role in this case, alongside precipitation, as the ISBA discharge anomaly is particularly pronounced. This point was not sufficiently emphasized in the text, so we have updated it:**

**« In mid-2018, the SSS anomalies reach ~0.3 pss (Figure 6a). As in 2011, all CROCO simulations, including those with climatological forcing (Figure 6b), reproduce this positive anomaly, which cannot be attributed to one particular forcing anomaly. Analysis of the salinity balance for CROCOimerg (Figure 7g) reveals that the salty anomaly initially results from a strong positive atmospheric forcing anomaly (i.e., rain deficit) in IMERG (Figure 7i), also found in the ERA5 product (Figure 6c). The greater impact of the IMERG precipitation anomaly on CROCOimerg SSS, compared to the impact of the ERA5 precipitation anomaly on CROCOisba SSS (Figure 6a) could be due to a more localized and intense precipitation anomaly in IMERG than in ERA5 (see Figure S.1). This precipitation anomaly is accompanied by a large ISBA runoff negative anomaly (Figure 6c, Figure 7g, black dashed curve), increasing SSS by means of a very large positive advection (Figure 7g, blue curve). This runoff anomaly explains why CROCOprclm (also forced by ISBA, Figure 6b) also simulates the positive SSS anomaly (albeit weaker than in CROCOimerg) without a precipitation anomaly. Because the GLOFAS runoff anomaly has an opposite sign (i.e. larger runoff), the CROCOglofas simulation displays a weaker SSS anomaly than the simulations forced by ISBA runoff (CROCOimerg and CROCOisba; Figure 6b). In the case of CROCOimerg (Figure 7g), the anomaly is due to both a rainfall and a runoff negative anomaly (yellow and blue curves).**
**In conclusion, the 2018 SSS anomaly is due to the combined effects of precipitation anomalies and river discharges. It is primarily caused by a strong precipitation negative anomaly (observed in both forcing datasets), which is not entirely compensated by entrainment (Figure 7g). This is then accompanied by a river discharge negative (positive) anomaly of ISBA (GloFAS) runoff, thereby accentuating (mitigating) the salinity anomaly through advection. This runoff anomaly explains the CROCOprclm SSS anomaly. The large GloFAS runoff is surprising as it is opposite to the rain deficit over the oceanic region during this period (Figure 6c). » (lines 445 - 463)**

C) There seems to have happened a mix-up in Figure 8 - the legends in the figure do not match the corresponding text which makes it rather hard to follow. It's not clear to me whether the text discussing Fig. 7a actually refers to Fig. 7b or whether just the legends in the Figure are wrong. Please correct and carefully check that legends, figure caption and corresponding text align.

**We thank the reviewer for pointing this. Panels a and b of Figure 8, and corresponding captions, were swapped in the previous response to the reviews; we have corrected this error.**

D) Specific comments:

- title: "…of Sea Surface Salinity in THE…"
- title and elsewhere: The region can be denoted as either "Eastern North Tropical Atlantic" or "Northeastern Tropical Atlantic" but not "East Northern"

**We thank the reviewer for his suggestion. The new title is "Influence of river runoffs and precipitation on the seasonal and interannual variability of Sea Surface Salinity in the Eastern North Tropical Atlantic."**

- line 13: "of the Intertropical…"

**agreed**

- line 14: "eastern part of the…"

**agreed**

- line 31: "The input of these low salinity waters lowers the density of the surface waters"

**agreed**

- line 48: ENSO has not been defined before

**Added definition of ENSO (line 47)**

- line 56: Not sure what is meant here. Are the variations expected to "increase by 10 to 28%" maybe?

**Yes, we reformulated as proposed**

- line 63: "strong" or "pronounced" instead of "thriving"

**We replaced « thriving » by « strong »**

- line 70/71: I guess this statement is meant for the region that is considered here?!?

**Yes, it is. We have reformulated as follows:**

**« The aforementioned studies demonstrate the usefulness of salinity as a tracer for variations in the water cycle, from the perspective of the seasonal cycle and interannual variability near major rivers. Concerning the impact of freshwater fluxes on the salinity in e-NTA region, only the seasonal variation and their driving physical processes have been studied by Camara et al. (2015) using the Nucleus for European Modelling of the Ocean (NEMO) ocean model. They found that runoffs and precipitations were the main contributors of the freshening in the e-NTA, and that poleward advection of low salinity waters along the coasts was partly compensated by vertical diffusion of salinity. However, to our knowledge, no study has yet focused on interannual variability in this region, nor on the sensitivity of the simulated salinity to the runoffs and precipitation forcing datasets. » (lines 64 - 71)**

**Instead of:**

**"Concerning the impact of freshwater fluxes on salinity in e-NTA region, only the seasonal variation and their driving physical processes have been studied by Camara et al. (2015) using the Nucleus for European Modelling of the Ocean (NEMO) ocean model. They found that runoffs and precipitations were the main contributors of the freshening in the e-NTA, and that poleward advection of low salinity waters along the coasts was partly compensated by vertical diffusion of salinity.
The aforementioned studies demonstrate the usefulness of salinity as a tracer for variations in the water cycle, both from the perspective of the seasonal cycle and interannual variability. However, to our knowledge, no study has yet focused on interannual variability."**

- line 71: replace the first "aim" with "goal"

**We have replaced the first « aim » by « goal ». Thanks.**

- Figure 1: Please specify the time period of the average (October to December of which years?) in the caption.

**We modified the caption: « CCI satellite SSS (color) averaged over October-November-December of years 2010 to 2019 (over ocean) »**

- line 152: either "at the open boundaries" or "as boundary open conditions"

**We replaced by « as open boundary conditions »**

- line 214: Which monsoon is meant here? Please explain in a bit more detail how this can be seen in Figure 8.

**We moved Figure 8 to this paragraph (as Figure 2) to simplify the reading, and we have rephrased:**

**"GloFAS and ISBA runoffs, after summing the individual outflows for the region studied, have similar climatologies (maximum difference of $1.10^8\,\mathrm{m^3/d}$, see Figure 3b). The simulated river runoffs exhibit strong interannual anomalies in this area (Figure 2). These river runoff anomalies are strongly correlated with African monsoon variations, as shown in Figure 2, where interannual anomalies of modelled runoffs closely mirror the interannual anomalies of precipitations over the watershed, used as forcing in these models. These interannual anomalies can reach $8.10^8\,\mathrm{m^3/d}$, i.e., almost 40 % of the seasonal variation (Figure 3b). They are sometimes of opposite signs between the two products, with differences reaching $1.10^9\,\mathrm{m^3/d}$ (Figure 2). These differences and their origins are discussed in section 4. » (lines 214 - 220)"**

**Instead of:**
**"GloFAS and ISBA runoffs, after summing the individual outflows for the region studied, have similar climatologies (maximum difference of $1.10^8\,\mathrm{m^3/d}$, see Figure 2b). The simulated river runoffs exhibit strong interannual anomalies in this area (Figure 5c). These river runoff anomalies are highly correlated with monsoon anomalies (Figure 8) and can reach $11\,000\,\mathrm{m^3s^{-1}}$, i.e., almost 30 % of the seasonal variation (Figure 2b). These interannual anomalies (see sect. 3.3 below) are sometimes of opposite signs between the two products, with differences reaching $12.10^8\,\mathrm{m^3/d}$ (Figure 5c). These differences and their origins are discussed in section 4."**

- line 343: What is meant by "the satellite data are slightly overestimated"?

**Satellites near the coast estimate a higher salinity than what is observed in situ on a few measurements. We made it clearer:**
**« Among the three types of gridded products, satellite observations show the closest alignment with in-situ data, with $r^2$ values of 0.94 and 0.89 when compared to Argo and TSG data, respectively (Table 2). The observed differences generally remain within 0.2 pss in absolute value (Figure 5b,f), except for a few instances involving in situ measurements taken very close to the coast.» (lines 344 - 347).**

**Instead of:**
**"Of the three types of gridded products, satellite observations are the closest to *in situ* data, with $r^2$ values of 0.94 and 0.89 for comparisons with Argo and TSG data respectively (Table 2). Except for TSG measurements taken very close to the coast where the satellite data are slightly overestimated, the differences observed rarely exceed 0.2 pss in absolute value (Figure 4b,f)."**

- Table 2: Does "global" really refer to the whole globe or just the full model domain? If really global is meant, what was used for the comparison with the model output outside of the model domain?

**"Global" refers to the full domain, we specified it in the table 2 caption.**

- line 453: probably Figure 6g instead of 6c
**Yes, this was a typo, we corrected it.**